# Lack of ANKMY2 suppresses kidney cystogenesis in embryonic- and adult-onset polycystic kidney disease

**Sun-Hee Hwang[1], Kyungsuk Choi[1], Hemant Badgandi[1¤], Kevin A. White[1], Yu Xun[1], Owen M. Woodward[2], Feng Qian[3], Saikat Mukhopadhyay[1]***

**1** Department of Cell Biology, University of Texas Southwestern Medical Center, Dallas, Texas, United States of America, **2** Departments of Physiology and Medicine, University of Maryland School of Medicine, Baltimore, Maryland, United States of America, **3** Division of Nephrology, University of Maryland School of Medicine, Baltimore, Maryland, United States of America

¤ Present address: Department of Biology and Environmental Science, University of New Haven, West Haven, Connecticut, United States of America.

\* saikat.mukhopadhyay@utsouthwestern.edu

## Abstract

Autosomal dominant polycystic kidney disease (ADPKD) is characterized by progressive bilateral cyst formation. Multiple cellular pathways including second messenger cAMP signaling are dysregulated in ADPKD, but mechanisms initiating cysts are unknown. ADPKD is caused by mutations in *PKD1*/*PKD2* genes encoding for polycystins that localize to primary cilia—nonmotile, microtubule-based dynamic compartments sensing extracellular chemical/mechanical signals. The compact cylindrical structure of cilia enables tunable signaling amplification regulatable by ciliary length. Severe cystogenesis from polycystin loss is cilia dependent and ciliary elongation is common in cystic epithelia. However, uncoupling the cilium-specific signals repressed by polycystins from downstream cystogenic pathways has proven challenging. Here we aim to understand roles of compartmentalized cAMP signaling in cystogenesis and ciliary length control. We investigated ANKMY2, an Ankyrin repeat MYND domain protein involved in maturation and ciliary localization of membrane adenylyl cyclases—enzymes generating cAMP. In kidney-specific *Ankmy2/Pkd1* knockout mice, loss of ANKMY2 suppressed early postnatal cystogenesis and significantly extended survival in an embryonic-onset *Pkd1* deletion model. Similarly, in an adult inducible *Pkd1* knockout model, ANKMY2 deficiency reduced cyst burden. Mechanistically, ANKMY2 controlled the ciliary trafficking of multiple adenylyl cyclases in mouse and human kidney epithelial cells without disrupting cilia while retaining cellular pools. Ciliary elongation began in dilatated tubules of adult onset ADPKD mice and further increased in cystic kidneys. Both initial and progressive phases of cilia lengthening were ANKMY2-dependent. Our findings indicate that ciliary adenylyl cyclase signaling likely promotes cilia-dependent cyst initiation distinct

**Data availability statement:** All relevant data are within the paper and its supporting information files.

**Funding:** This study was supported by the National Institutes of Health grant R01DK128089 (SM). The content is solely the responsibility of the authors and does not necessarily represent the official views of the National Institutes of Health. The funders had no role in study design, data collection and analysis, decision to publish, or preparation of the manuscript.

**Competing interests:** The authors have declared that no competing interests exist.

from cyst progression involving cellular cAMP. Importantly, kidneys lacking ANKMY2 did not show ciliary elongation despite elevated cAMP, suggesting that cilia lengthening during cyst progression could be contingent upon pre-cystic ciliary regulation. These results suggest a critical role for compartmentalized adenylyl cyclase signaling in ADPKD pathogenesis and a framework for identifying ciliary effectors and early subcellular events in cystogenesis.

## Author summary

Abnormal formation of fluid sacs called cysts occurs in kidney nephrons in polycystic kidney disease (PKD). Increased cellular levels of the second messenger cAMP occurs in cysts. Current therapies are primarily designed to antagonize cAMP increase in cysts but are not fully effective in reducing cyst burden, highlighting the need to better understand the mechanisms underlying cyst initiation and progression. Polycystin channels, the major gene products affected in PKD, localize to primary cilia. Cilia are minute cellular compartments that effectively regulate cellular signaling. Cyst formation occurring from polycystin loss requires cilia, suggesting that polycystins normally inhibit cyst promoting signals within cilia. Identifying these cilia localized signals could reveal effective therapeutic targets. Here we studied an Ankyrin repeat and MYND domain protein, ANKMY2 that determines maturation and ciliary localization of adenylyl cyclases—enzymes generating cAMP. We found that loss of ANKMY2 suppresses cyst formation in both early postnatal and adult mouse models of PKD. Our results suggest that ANKMY2-dependent trafficking of adenylyl cyclases to cilia likely promotes compartmentalized ciliary adenylyl cyclase signaling, initiating cyst formation through a mechanism distinct from later cyst progression involving cellular cAMP levels. Targeting ciliary-specific adenylyl cyclase function could improve current therapies that reduce cellular cAMP levels.

## Introduction

The kidney nephron tubular epithelial cells have primary cilia starting from development and persisting throughout adult life [1]. The primary cilium is a paradigmatic subcellular compartment for generating signaling outputs that can have profound effects on cellular function [2]. The role of primary cilia in kidney tubule morphogenesis and in the related disease such as polycystic kidney disease is complex [3,4] and is also developmental stage-dependent [5]. Genetic epistasis studies—using cilia loss in the context of polycystic kidney disease—have suggested that primary cilia generate both positive and negative counterregulatory signals that modulate the severity of cystogenesis [3,4]. Furthermore, distinct temporal postnatal developmental windows of polycystin function have been shown to determine the timing of cyst initiation, with polycystin loss before P14 driving early cystogenesis [5]. Cyst

progression is regulated by a multifactorial and multi-organellar process [6], thus studying the precise contribution of cilia in cystogenesis has been complicated. Furthermore, cilia localized proteins have extraciliary components and teasing apart ciliary from extraciliary function from knockout studies is not feasible [7]. As lack of cilia prevents studying signals that are generated by cilia, preventing ciliary compartmentalization without disrupting the organelle is needed to study ciliary contributions in downstream pathways [7].

ADPKD is caused by mutations in genes (*Pkd1* and *Pkd2*) encoding polycystin-1 (PC1) and polycystin-2 (PC2)—both of which localize to primary cilia [8,9]. Loss of polycystins causes severe cystogenesis, which is mostly cilia dependent, suggesting that such loss derepresses cyst activators [10]. However, the polycystin-inhibited cilia-dependent/localized cyst activator(s) (CDCA) that promote cyst growth in ADPKD are unknown. The CDCA should in principle localize to cilia, should not affect cystogenesis on being depleted but should prevent cystogenesis in embryonic- or adult-onset cystogenesis upon concomitant loss in the background of PC1/2 depletion [4]. A clue to the positive signal(s) came from studying TULP3, a ciliary trafficking adapter pivotal in ciliary membrane composition that functions in coordination with the IFT-A complex [11]. Deletion of *Tulp3* and IFT-A subunit *Ift139* in adult-onset disease causes inhibition of cystogenesis [12,13], whereas deletion of *Tulp3* exacerbates early postnatal cystogenesis from *Pkd1/2* loss [14].The differences in genetic epistasis between *Tulp3* and *Pkd1* between early- and adult-onset disease [4] is explained by TULP3 mediated trafficking of multiple cargoes to the ciliary membrane [11]. For example, the exacerbation of early-onset disease from TULP3 loss is at least partly from lack of trafficking of ARL13B, a bona fide TULP3 cargo, to cilia [15–17]. The ciliary components of CDCA signal(s) in ADPKD cystogenesis remain unknown, although cytoplasmic components downstream of cilia, such as cell cycle kinases [18], the transcription factor GLIS2 [19], and the ankyrin- repeat protein ANKS3 [20], have recently been described.

Previous studies have demonstrated that cellular cAMP is a major driver of cystogenesis in ADPKD [21] and in other cystic kidney disease models including ARPKD [22], NPHP [23] and juvenile cystic kidney disease from loss of NEK8 [24]. Stimulation by cAMP analogues such as 8-Br-cAMP has also been shown to contribute to cyst growth by stimulating fluid secretion [25] through activation of the CFTR chloride channel [26,27] and by increasing cell proliferation through activation of the B-Raf/MEK/ERK pathway [28]. Conversely, drugs that reduce intracellular cAMP levels, such as vasopressin receptor 2 (V2R) antagonists and somatostatin receptor agonists, inhibit cyst growth in ADPKD and other cystic kidney disease mice models [23]. Together, these drugs reduce cellular cAMP levels, reduce cystic burden in animal models [29] and have renoprotective effects in ADPKD patients [30]. However, distinguishing ciliary adenylyl cyclase signaling from cellular cAMP dependent processes in cystogenesis have been challenging. Achieving spatial specificity in targeting ciliary signaling could help uncover the cystogenic signals normally repressed by polycystins

The tiny cylindrical nature of cilium (up to 5 µm long, 200 nm in diameter) enriches and amplifies signaling reactions that can be tuned by the dynamic regulation of ciliary axoneme in controlling steady-state length [31–33]. Ciliary length is exquisitely controlled across tissues and species by factors that localize in the cilia-centrosomal complex [34,35]. Loss of PC1/2 increases cilia length in ADPKD mice models [13,36–38] and in human patients [38]. although there are conflicting results in cell lines showing both increased [39] or decreased cilia lengths [40]. Increased *Pkd1* dosage also induces increased cilia length in both pre-cystic and cystic tubules [41], suggesting that polycystin levels modulate cilia length in a context-dependent manner.

Importantly, cAMP has been linked with ciliary length regulation. Elevated cellular cAMP levels acutely increases ciliary length in cultured kidney epithelial cells [42], and optogenetic elevation of ciliary cAMP also results in ciliary elongation and cystic changes in 3D cultured cells [43,44]. However, because of miniscule volume of cilia compared to the cytoplasm–less than 1/30000 of the total cellular volume [33]–any changes in ciliary cAMP are unlikely to cause global cAMP changes [31,32]. Thus, high cAMP levels in cystic epithelia are unlikely due to diffusion of ciliary cAMP into the cytoplasm. Despite these associations, the pathophysiological role of ciliary length regulation in cystogenesis remains unclear. Whether ciliary elongation is a cause or consequence of cyst formation has not been examined.

Membrane adenylyl cyclases are topologically complex enzymes having two transmembrane bundles of 6 transmembrane domains each that bring together two pseudosymmetric cyclase domains catalyzing production of cAMP from ATP [45]. At least three of the nine membrane adenylyl cyclases traffic to cilia, including ADCY3, ADCY5, and ADCY6 [46–48]. The role of membrane adenylyl cyclases ADCY5 and ADCY6 was previously tested separately in PKD models using global knockout [49] or conditional knockout strategies [50], respectively, which were shown to have partial amelioration of cystic burden. Knockouts or conditional knockouts of *Adcy5* and *Adcy6* also reduce kidney cAMP levels [49,50]. ADCY3 has been shown to be expressed in human ADPKD kidney cysts [51]. Downregulation of the cAMP effector protein kinase A (PKA) using kidney-specific expression of a dominant negative PKA regulatory subunit *RIaB* allele delays development and progression of PKD in *Pkd1$^{RC/RC}$* models [52]. The promotion of cystogenesis by PKA in ADPKD mice models is also through modulation of cellular cAMP levels [52]. Correspondingly, kidney-specific knockout of PKA regulatory subunit *RIa* upregulates PKA activity, induces cystic disease in wild-type mice and aggravates it in *Pkd1$^{RC/RC}$* mice [53]. Whereas cAMP levels increase with severity of cystic disease [54], these studies do not account for the ciliary component of cAMP signaling in pathogenesis of ADPKD. Some membrane adenylyl cyclases have been shown to be bona fide IFT-A cargoes to cilia [55]. However, dissecting specific role of ciliary compartmentalization of multiple adenylyl cyclases while still retaining cellular pools and without disrupting cilia has been challenging.

In this study, we employ a targeted approach to selectively study adenylyl cyclase signaling from cilia in mouse ADPKD models. We generated nephron-specific conditional knockout mice for ANKMY2, an Ankyrin repeat and MYND domain containing protein we recently discovered as a repressor of hedgehog signaling during mouse neural tube development [56]. ANKMY2 determines maturation and trafficking of multiple adenylyl cyclases to cilia without depleting their overall cellular levels [56], unlike adenylyl cyclase knockouts [49,50]. Importantly, high hedgehog signaling in the *Ankmy2* knockout mouse neural tube was fully cilia-dependent [56], suggesting dysregulated ciliary/peri-ciliary adenylyl cyclase signaling from ANKMY2 loss. Here we demonstrate that ANKMY2 mediated adenylyl cyclase targeting to renal epithelial cilia and loss of ANKMY2 suppressed cystogenesis in both embryonic- and adult-onset ADPKD models. ANKMY2 deficiency prevented the ciliary elongation normally observed in *Pkd1*-deficient renal epithelial cells. Cilia lengthening preceded cyst formation in an ANKMY2-dependent manner and kidneys lacking ANKMY2 did not show ciliary elongation despite elevated cAMP levels, suggesting ciliary lengthening during cyst progression could be contingent upon pre-cystic regulation. These results suggest that ANKMY2-mediated ciliary adenylyl cyclase signaling could initiate cilia-dependent cyst activation from coincidental loss of polycystins. The temporal windows of compartmentalized ciliary signaling and ciliary length regulation that we identify will enable devising ADPKD treatment strategies.

## Results

### Genetic ablation of ANKMY2 suppresses early cystogenesis in embryonic-onset PKD

We previously demonstrated high hedgehog signaling in the mouse embryonic neural tube in the *Ankmy2* knockout that was fully cilia-dependent from loss of targeting of multiple adenylyl cyclases to cilia [56]. Previous studies have tested the role of ADCY5 and ADCY6 separately in PKD models using straight knockout [49] or conditional knockout strategies [50], respectively. Lack of ANKMY2 allowed us to test the specific role of multiple adenylyl cyclases without generating knockouts of multiple adenylyl cyclase while still retaining cellular pools. However, as *Ankmy2* knockouts are E8.5 lethal, we generated a conditional knockout allele of *Ankmy2* [56]. Using the collecting duct-specific *HoxB7-Cre* that starts expressing in the embryonic kidney, we generated conditional knockouts of *Ankmy2* along with *Pkd1*. By P3, the *HoxB7-Cre*; *Pkd1$^{f/f}$* animals had significantly increased 2-kidney/body weight and kidney cystic index. Notably, *HoxB7-Cre*; *Pkd1$^{f/f}$*; *Ankmy2$^{f/f}$* animals showed a significant reduction in kidney to body weight and cystic index compared to *HoxB7-Cre*; *Pkd1$^{f/f}$* animals (Fig 1A–1C). Hematoxylin and Eosin (H&E) staining for all kidneys assessed are shown in S1A Fig. Immunostaining of kidney sections confirmed cysts in AQP2+collecting ducts in *HoxB7-Cre*; *Pkd1$^{f/f}$* that was rescued in *HoxB7-Cre*; *Pkd1$^{f/f;}$ Ankmy2$^{f/f}$* mice (Fig 1D–1E). The *HoxB7-Cre*; *Ankmy2$^{f/f}$* animals were not affected (Figs 1A–1E and

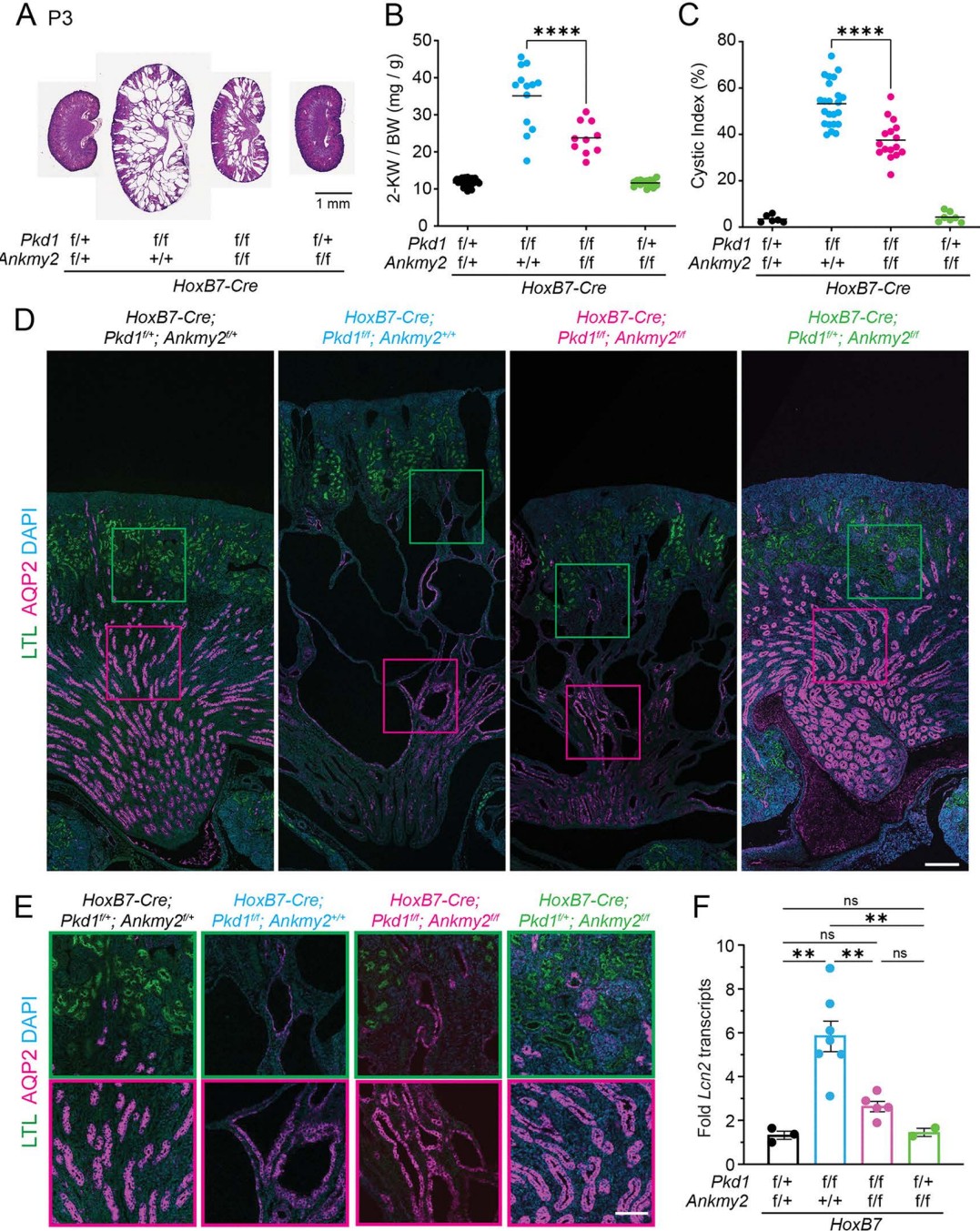

**Fig 1. Early cystogenesis in embryonic-onset PKD is suppressed from lack of ANKMY2. (A)** H&E images of kidneys in control, *HoxB7-Cre; Pkd1^fl/fl*, *HoxB7-Cre; Pkd1^fl/fl; Ankmy2^fl/fl*, *HoxB7-Cre; Ankmy2^fl/fl* mice at P3 in C57BL/6J background. Images of multiple kidneys shown in S1A Fig. **(B)** 2-Kidney to body weight ratios at P3 in *HoxB7-Cre; Pkd1^fl/fl* mice were significantly higher than *HoxB7-Cre; Pkd1^fl/fl; Ankmy2^fl*. P3 animals had body weight between 2.5-3 g. Horizontal bars, mean. **(C)** Significantly increased cystic index in *HoxB7-Cre; Pkd1^fl/fl* mice compared to *HoxB7-Cre; Pkd1^fl/fl; Ankmy2^fl/fl*. Horizontal bars, mean. **(D-E)** AQP2/LTL co-staining in kidney sections at P3 shows multiple Aqp2 positive cysts in medullar and corticomedullar junctions in *HoxB7-Cre; Pkd1^fl/fl* animals compared to *HoxB7-Cre; Pkd1^fl/fl; Ankmy2^fl/fl*. Scale: 200 μm. Magnified images are shown in **E**. LTL positive, green outlines; AQP2 positive, magenta outlines. Scale: 200 μm **(D)**, 100 μm **(E)**. **(F)** Transcript levels of *Lnc2* in whole kidneys from *HoxB7-Cre; Pkd1^fl/fl* mice are significantly higher than *HoxB7-Cre; Pkd1^fl/fl; Ankmy2^fl/fl*. Data are shown as mean±SEM. ****, P<0.0001; **, P<0.01; ns, not significant using one-way ANOVA with Sidak's multiple comparisons tests **(B, C, F)**. See also S1 Fig.

S1A). Transcript levels of *Pkd1* and *Ankmy2* collected from total kidneys trended towards being reduced in corresponding conditional knockouts, respectively, compared to controls (S1B Fig). As epithelial cells in nephrons at this early postnatal stage are still proliferating, we tested transcript levels of *Lipocalin-2* (*LCN2*) as an orthogonal measure of cystogenesis. LCN2, also known as neutrophil gelatinase-associated lipocalin (NGAL), is associated with cyst expansion in PKD in rodent models and humans [57]. Notably, levels of *Lcn2* transcripts significantly increased in *Hoxb7-Cre; Pkd1^f/f* cystic kidneys compared to *Hoxb7-Cre; Pkd1^f/f; Ankmy2^f/f* P3 kidneys (Fig 1F). Thus, early cystogenesis in embryonic-onset PKD is suppressed from lack of ANKMY2.

## ANKMY2 loss improves survival in embryonic-onset PKD without affecting later cystic burden

As early postnatal cystogenesis in *HoxB7-Cre; Pkd1^f/f* was ANKMY2-dependent, we next checked the life expectancy and progressive cystic burden of these mice. Interestingly, the life expectancy of *HoxB7-Cre; Pkd1^f/f* animals was significantly less compared to *HoxB7-Cre; Pkd1^f/f; Ankmy2^f/f* animals, whereas *HoxB7-Cre; Ankmy2^f/f* animals remained unaffected similar to controls (Fig 2A). However, at P14-P15, the 2-kidney to body weight and cystic index of the kidneys were not significantly different between *HoxB7-Cre; Pkd1^f/f* and *HoxB7-Cre; Pkd1^f/f; Ankmy2^f/f* animals irrespective of sex (Fig 2B–2D). H&E staining for all kidneys assessed are shown in S2 Fig. We next tested if cAMP-dependent phosphorylation of CREB is affected from loss of ANKMY2. The levels of pCREB in nucleus are reflective of cellular cAMP levels that activates protein kinase A [58,59]. We noted robust nuclear localization of pCREB by immunofluorescence in *HoxB7-Cre; Pkd1^f/f* mice in postnatal kidneys at P15. The nuclear localization of pCREB at P15 was also similar between *Hoxb7-Cre; Pkd1^f/f; Ankmy2^f/f* and *Hoxb7-Cre; Pkd1^f/f* animals (Fig 2E). These results suggested that total cellular cAMP level responses remained unaffected from *Ankmy2* deletion at this later stage with similar cystic burden. Therefore, lack of ANKMY2 suppressed early cystogenesis in embryonic-onset PKD and improved life expectancy with no effect on later cystic burden.

## Genetic ablation of ANKMY2 suppresses cystogenesis in adult-onset PKD

We next tested the role of ANKMY2 in adult-onset PKD models. We utilized the pan nephron-specific deletion using the doxycycline inducible *Pax8^rtTA; TetO-Cre* digenic system to delete *Pkd1* and/or *Ankmy2* using conditional alleles. We injected Doxycycline intraperitoneally at P28-30 and performed analysis at 5 months of age (Fig 3A). qRT-PCR of whole kidneys showed reduction of *Pkd1* or *Ankmy2* in the respective conditional knockout animals at 5 months (S3A Fig). At this age, *Pax8^rtTA; TetO-Cre; Pkd1^f/f* (abbreviated as *Pkd1 cko*) mice showed increased 2-kidney to body weight ratio and cystic index compared to controls (Figs 3B–3D **and** S3B). We noted a significant reduction in kidney to body weight ratio in *Pax8^rtTA; TetO-Cre; Pkd1^f/f; Ankmy2^f/f* (abbreviated as *Dko*) compared to *Pkd1 cko* male animals. In addition, the cystic index was significantly reduced in the *Dko* compared to *Pkd1 cko* mice (Fig 3C–3D). *Pax8^rtTA; TetO-Cre; Ankmy2^f/f* kidneys remained unaffected. Immunostaining of kidney sections confirmed cysts in both LTL+ proximal tubules and AQP2+ collecting ducts in *Pkd1 cko* that was rescued in *Dko* male mice (Figs 3E–3F **and** S3B). Cyst sizes in *Pkd1 cko* were impacted in both LTL+ and AQP2+ cysts compared to *Dko.* The BUN values *Pkd1 cko* was significantly higher than *Dko* male mice, and the levels correlated with cyst severity (Fig 3G). However, kidney to body weight ratio, cystic index and cyst size of both LTL+ and AQP2+ cysts were not significantly different between *Pkd1 cko* and *Dko* female mice (S4A-D Fig), unlike in males. The lack of effect in female in contrast to male mice suggested sex-specificity in effects of ANKMY2 in the PKD model most likely from androgen receptor mediated control of sexually dimorphic gene expression in tubule segments [60,61]. Overall, lack of ANKMY2 suppressed adult onset cystogenesis in male mice.

## Increased proliferation in adult-onset PKD is suppressed by loss of ANKMY2

We next tested proliferation in the adult-onset PKD models. We immunostained kidney sections for Ki67 and quantified the percentage of cycling cells by counting Ki67+ epithelial cells co-stained with LTL (proximal tubule) or AQP2 (collecting duct). For this quantification, we tested kidneys from male mice with cystic indices close to the median values of

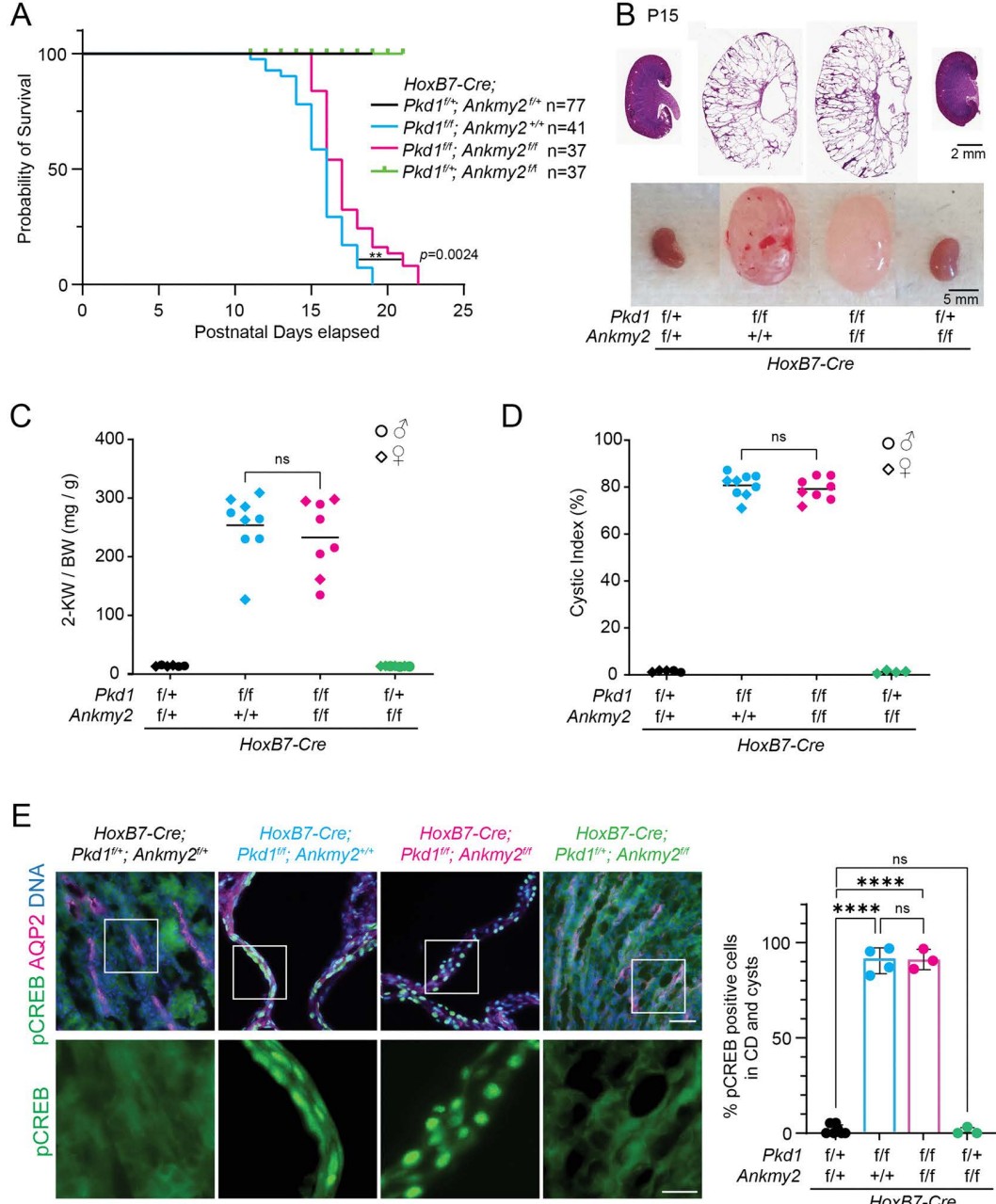

**Fig 2. Lack of ANKMY2 improves life expectancy in embryonic-onset PKD despite having no effect on later cystic burden. (A)** Kaplan-Meier survival curves of control (N=77), *HoxB7-Cre; Pkd1^f/f* (N=41), *HoxB7-Cre; Pkd1^f/f; Ankmy2^f/f* (N=37) and *HoxB7-Cre; Ankmy2^f/f* (N=37) mice shown by postnatal days elapsed. **, P=0.0024 by log-rank (Mantel-Cox) test. **(B)** H&E and whole mount images of P15 kidneys in control, *HoxB7-Cre; Pkd1^f/f*, *HoxB7-Cre; Pkd1^f/f; Ankmy2^f/f*, *HoxB7-Cre; Ankmy2^f/f* mice at P15 in C57BL/6J background. Images of multiple kidneys shown in S2 Fig. **(C)** 2-Kidney to body weight ratios at P15 in *HoxB7-Cre; Pkd1^f/f* mice were not significantly different from *HoxB7-Cre; Pkd1^f/f; Ankmy2^f/f*. Both males (circles) and females (rhombi) are shown. Horizontal bars, mean. **(D)** At P15 cystic index in *HoxB7-Cre; Pkd1^f/f* kidneys compared to *HoxB7-Cre; Pkd1^f/f; Ankmy2^f/f* were not significantly different. Both males (circles) and females (rhombi) are shown. Horizontal bars, mean. **(E)** AQP2/pCREB co-staining in kidney sections at P15 shows comparable nuclear pCREB levels in *HoxB7-Cre; Pkd1^f/f* animals and *HoxB7-Cre; Pkd1^f/f; Ankmy2^f/f* cysts. Scale: Upper panels, 50 μm; Magnified lower panels, 20 μm. Quantification from multiple kidneys shown to the right. Data are shown as mean±SD. ****, P<0.0001; ns, not significant using one-way ANOVA with Sidak's multiple comparisons tests **(C, D, E)**. See also S2 Fig.

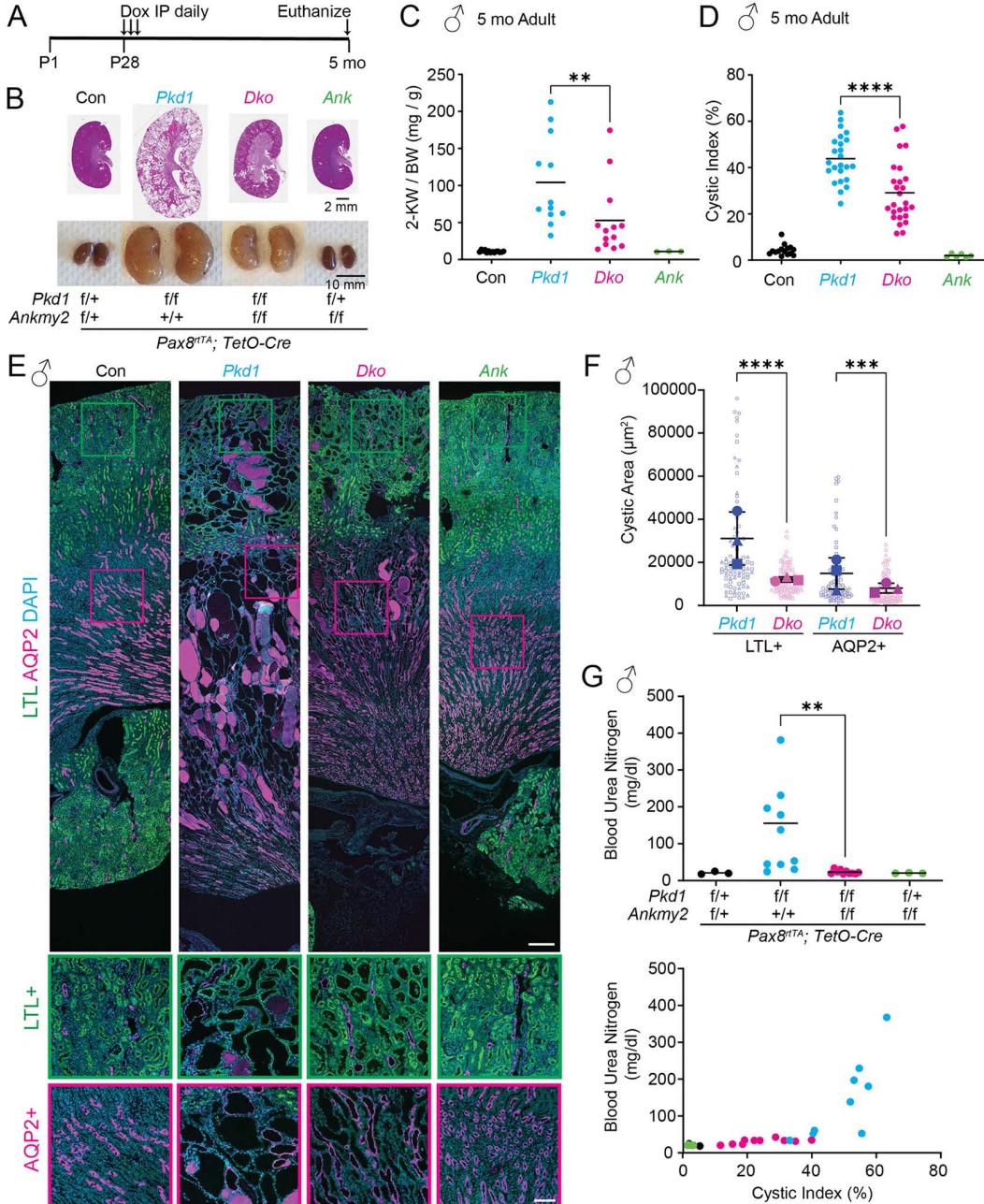

**Fig 3. Cystogenesis in adult-onset PKD in male mice is suppressed from lack of ANKMY2. (A)** Scheme for inducible conditional knockout in kidney nephron epithelia in an adult-onset model of PKD. **(B)** H&E and whole mount images of 5-month (mo)-old kidneys in control, $Pax8^{rtTA}$; $TetO$-$Cre$; $Pkd1^{f/f}$, $Pax8^{rtTA}$; $TetO$-$Cre$; $Pkd1^{f/f}$; $Ankmy2^{f/f}$, $Pax8^{rtTA}$; $TetO$-$Cre$; $Ankmy2^{f/f}$ male mice in C57BL/6J background. Images of multiple kidneys in male mice are shown in S3 Fig. **(C)** 2-Kidney to body weight ratios of 5-month-old $Pax8^{rtTA}$; $TetO$-$Cre$; $Pkd1^{f/f}$ male mice were significantly higher than $Pax8^{rtTA}$; $TetO$-$Cre$; $Pkd1^{f/f}$; $Ankmy2^{f/f}$. Horizontal bars, mean. **(D)** Significantly increased cystic index in $Pax8^{rtTA}$; $TetO$-$Cre$; $Pkd1^{f/f}$ male mice compared to $Pax8^{rtTA}$; $TetO$-$Cre$; $Pkd1^{f/f}$; $Ankmy2^{f/f}$. Horizontal bars, mean. **(E)** AQP2/LTL co-staining in kidney sections at 5 months shows multiple LTL positive cysts in cortical regions and corticomedullar junctions and AQP2 positive cysts in medullar, corticomedullar junctions and cortical regions in $Pax8^{rtTA}$; $TetO$-$Cre$; $Pkd1^{f/f}$ animals compared to $Pax8^{rtTA}$; $TetO$-$Cre$; $Pkd1^{f/f}$; $Ankmy2^{f/f}$. Scale: 300 µm. Magnified images of cortical and cortico-medullar junctions are shown below; LTL positive, green outlines; AQP2 positive, magenta outlines. Scale: 100 µm. **(F)** Cyst sizes in $Pax8^{rtTA}$; $TetO$-$Cre$; $Pkd1^{f/f}$ were increased in both LTL+ and AQP2+ cysts compared to $Pax8^{rtTA}$; $TetO$-$Cre$; $Pkd1^{f/f}$; $Ankmy2^{f/f}$ but more so in LTL+ proximal tubule cysts, which showed more differences between the two genotypes. Superplots of N = 3 kidneys/genotype are shown with cysts from each kidney depicted by smaller identical shapes and averages per kidney by analogous larger shapes. Cystic indices in $Pax8^{rtTA}$; $TetO$-$Cre$; $Pkd1^{f/f}$ were 63, 47, and 40, whereas that in $Pax8^{rtTA}$; $TetO$-$Cre$; $Pkd1^{f/f}$; $Ankmy2^{f/f}$

were 22, 15, and 18. Data are shown as mean±SD. **(G)** The BUN values of *Pax8^rtTA*; *TetO-Cre*; *Pkd1^flf* was significantly higher than *Pax8^rtTA*; *TetO-Cre*; *Pkd1^flf*; *Ankmy2^flf* mice, and the levels correlated with cystic index across genotypes. Horizontal bars, mean. ****, P<0.0001; ***, P<0.001; **, P<0.01 using one-way ANOVA with Sidak's multiple comparisons tests **(C, D, F, G)**. See also S3 and S4 Figs.

respective genotypes. Ki67 + cyclic cells in proximal tubule and collecting duct epithelia were significantly increased in *Pkd1 cko* compared to *Dko* (Fig 4A–4C). Cyst formation in conditional knockouts of *Pkd1/2* is associated with activation of components of the extracellular regulated kinase (ERK) pathway [10,62]. The amounts of phosphorylated and total ERK in whole kidney lysates increased in *Pkd1 cko* cystic kidneys compared to *Dko* kidneys (Fig 4D). Finally, as an orthogonal measure of cyst severity, we checked the transcript levels of Kidney Injury Molecule 1 (KIM-1). KIM1 is a transmembrane glycoprotein that is expressed mostly in proximal tubular cells and is elevated early in tubulointerstitial injury in both rodent models [63,64] and in humans [65]. Levels of *Kim1* transcripts significantly increased in *Pkd1 cko* cystic kidneys compared to *Dko* kidneys and remained unchanged in *Pax8^rtTA*; *TetO-Cre*; *Ankmy2^flf* kidneys compared to controls (Fig 4E). These results suggest that cyst burden is suppressed in adult-onset PKD from lack of ANKMY2 from decreased proliferation, reduced ERK activation, and exhibits reduced expression of tubulointerstitial injury marker KIM-1.

## Loss of ANKMY2 reduces the ciliary localization of adenylyl cyclases ADCY5 and ADCY6 in kidney epithelial cells

We next determined the mechanism by which ANKMY2 was regulating PKD cystogenesis. We previously showed that lack of ANKMY2 prevents localization of stably expressed ADCY3/5/6 in cilia of mouse fibroblasts and in neuroepithelial cells without grossly affecting cilia architecture [56]. Of the nine membrane adenylyl cyclases, ADCY6 is the predominant adenylyl cyclase detected throughout the nephron, whereas ADCY5 is the second most prevalent adenylyl cyclase detected in the connecting tubule and cortical collecting duct using proteomic studies of the nephron segments [66]. As ADCY5 and ADCY6 are expressed in kidney epithelia, we stably expressed LAP-tagged versions in mouse IMCD3-FlpIn cells (LAP N-term tag: GFP-TEV-S tag-ADCY; LAP-C-tern tag: ADCY-S tag-PreScission-GFP). Both ADCY5^LAP and ^LAPADCY6 localized to IMCD3-FlpIn cilia, although ^LAPADCY6 positive ciliary percentage was less than in the previously reported NIH 3T3 cells [56] (Fig 5A–5D). In addition, both these adenylyl cyclases were also localized to the secretory pathway and plasma membranes (S5A-S5B Fig). We next generated *Ankmy2* knockouts in the ADCY5^LAP and ^LAPADCY6 IMCD3 cells that was confirmed from the complete loss of ANKMY2 using immunoblotting for ANKMY2 (Figs 5G **and** S5C). Interestingly, we noted a significant decrease of ADCY5^LAP and ^LAPADCY6 ciliary and plasma membrane levels in *Ankmy2* ko cells despite persistent extraciliary localization in the secretory pathway (Figs 5A–5D **and** S5A-S5B). Gross ciliary morphology or localization of other TULP3 cargoes such as GPR161 or other non-TULP3 trafficked cilia localized proteins such as the hedgehog pathway transducer—SMO remained unaffected in *Ankmy2* ko IMCD3 cells [56]. We previously showed that complex glycosylation of adenylyl cyclases were affected in *Ankmy2* ko fibroblasts [56]. Both ADCY5 and ADCY6 were in complex glycosylated forms in IMCD3 cells, as determined by EndoH and PNGase treatments (Fig 5E– 5F). The levels of complex glycosylated forms of ADCY5/6 were reduced in *Ankmy2* ko IMCD3-FlpIn cells despite equivalent expression but were restored upon stable reexpression of ^HAANKMY2 (Fig 5G). Such reexpression also restored plasma membrane levels of ADCY5^LAP and ^LAPADCY6 (S5A-S5B Fig). Although reexpression did not rescue ciliary levels of ADCY5^LAP and ^LAPADCY6 in IMCD3 cells, we previously showed that ciliary levels were partially restored in mouse fibroblasts [56]. Such lack of rescue of ciliary levels is likely from extensive plasma membrane localization in IMCD3-FlpIn vs 3T3-FlpIn cells. Thus, lack of ANKMY2 prevents ciliary localization of ADCY5^LAP and ^LAPADCY6 while still retaining cellular levels. In contrast, in the *Adcy5/6* knockouts, total cellular levels of respective adenylyl cyclases would be absent. Like others, we were unable to specifically detect endogenous ADCY5/6 in the kidney by immunostaining with available antibodies [47,49,50,66]. However, our data from renal epithelial cell lines combined with nephron segment-specific proteomic profiles [66] suggest that conditional knockouts of *Ankmy2* likely prevented ciliary localization of multiple adenylyl cyclases in the kidney tubules.

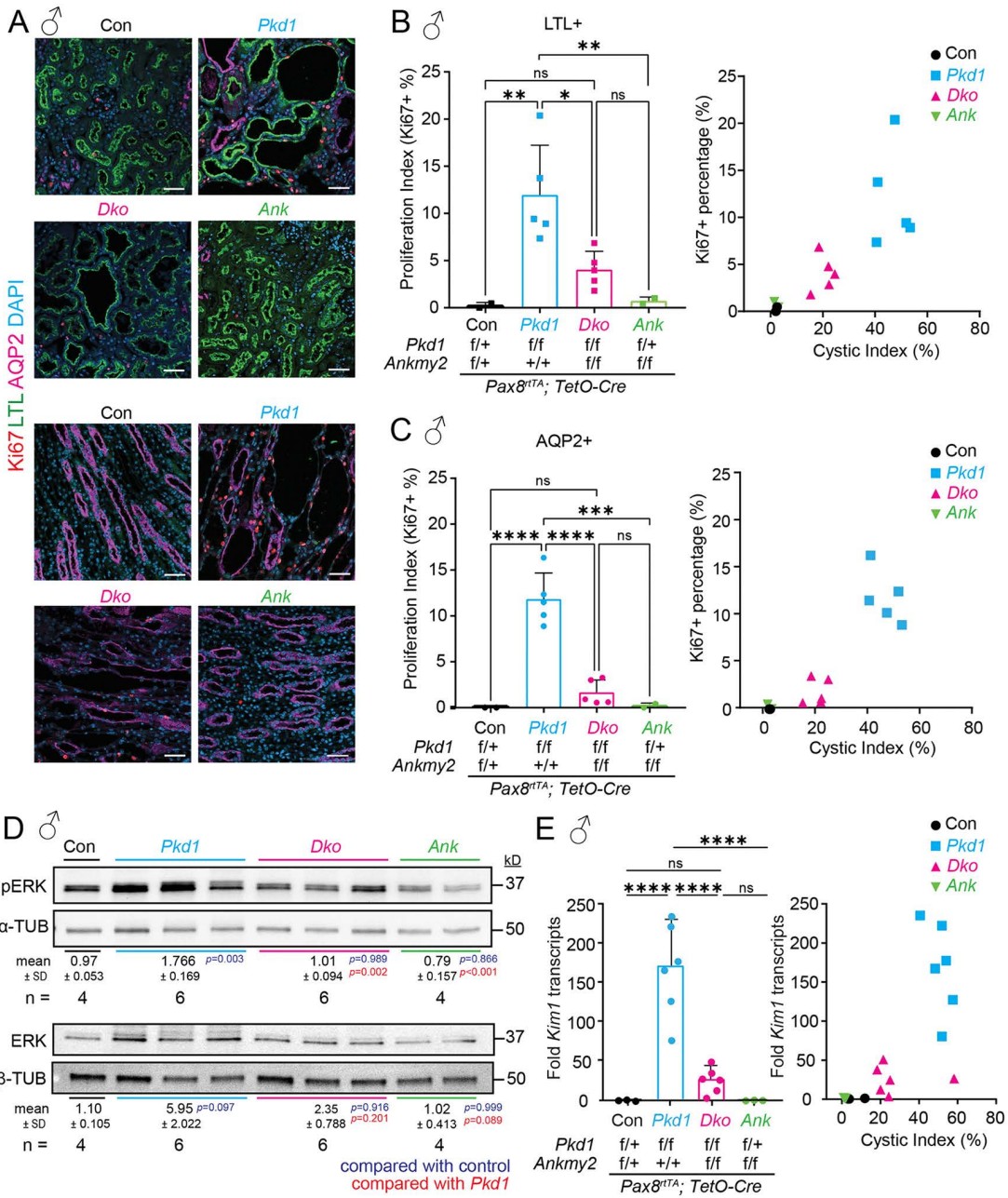

**Fig 4. Increased proliferation in adult-onset PKD in male mice is suppressed from lack of ANKMY2. (A-C)** Increased proliferation in cystic kidneys in 5-month-old *Pax8^rtTA*; *TetO-Cre*; *Pkd1^f/f* male mice compared to *Pax8^rtTA*; *TetO-Cre*; *Pkd1^f/f*; *Ankmy2^f/f*. Kidney sections were immunostained for Ki67, AQP2 and LTL and counterstained with DAPI. Ki67+cells were quantified. Regression analysis with respect to cystic index is shown. Scale: 50 μm. **(D)** Immunoblotting for pERK and ERK, of **(i)** control, *Pax8^rtTA*; **(ii)** *TetO-Cre*; *Pkd1^f/f*, *Pax8^rtTA*; *TetO-Cre*; **(iii)** *Pkd1^f/f*; *Ankmy2^f/f* and *Pax8^rtTA*; **(iv)** *TetO-Cre*; *Ankmy2^f/f* kidneys from male mice at 5 months. Loading controls (Tubulin) for each phosphoprotein is shown. Levels of individual phosphoprotein or total protein, each normalized to tubulin separately, is shown. Control, N=4; *Pax8^rtTA*; *TetO-Cre*; *Pkd1^f/f* N=6; *Pkd1^f/f*; *Ankmy2^f/f* and *Pax8^rtTA* N=6; and *TetO-Cre*; *Ankmy2^f/f* N=4. **(E)** Transcript levels of *Kim1* in whole kidneys from *Pax8^rtTA*; *TetO-Cre*; *Pkd1^f/f* are significantly higher than *Pax8^rtTA*; *TetO-Cre*; *Pkd1^f/f*; *Ankmy2^f/f*, and the levels correlated with cystic index across genotypes. Data are shown as mean±SD **(B-E)**. ****, P<0.0001; ***, P<0.001; **, P<0.01; *, P<0.05; ns, non-significant using one-way ANOVA with Sidak's multiple comparisons tests **(B-E)**.

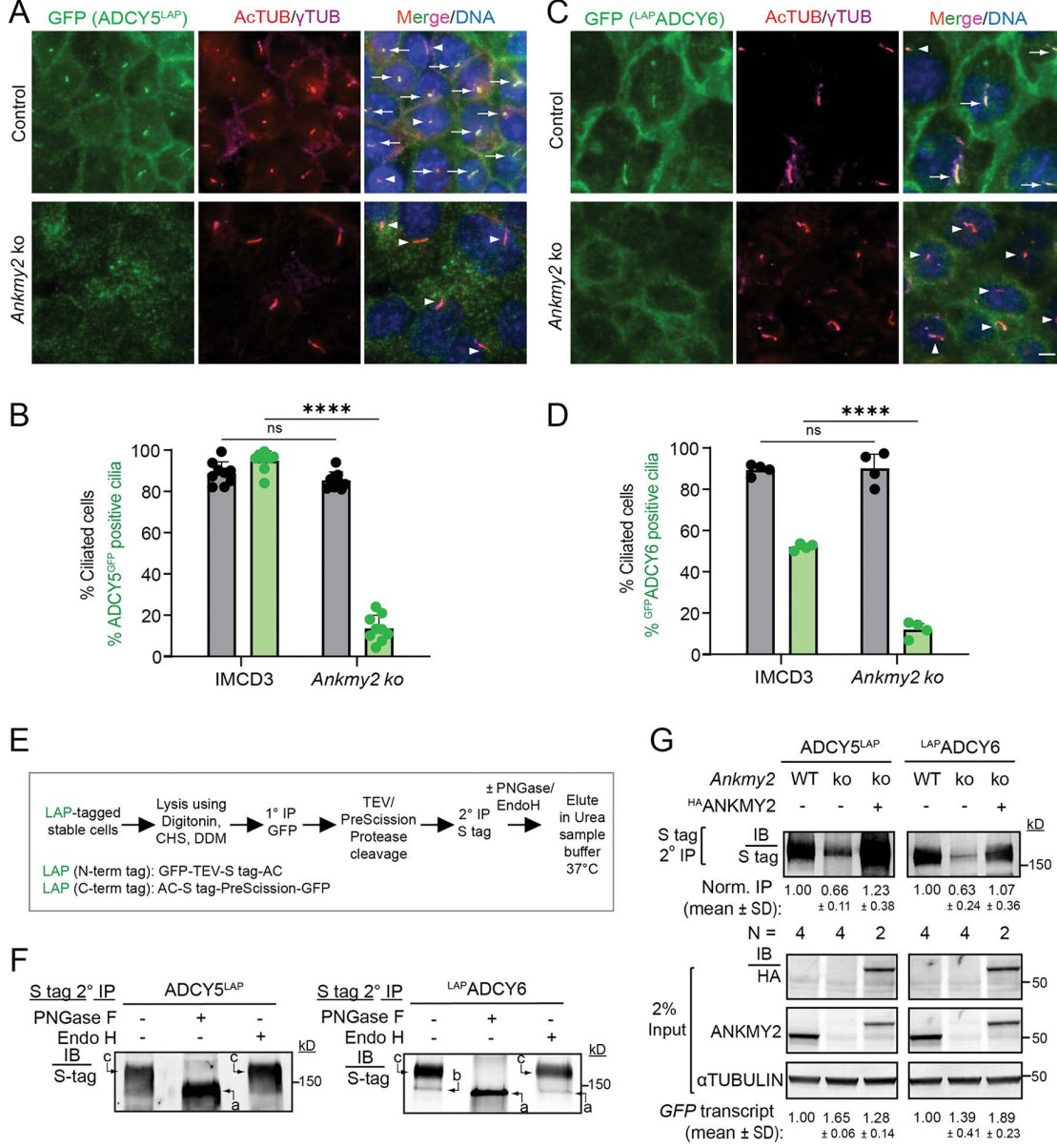

**Fig 5. Ciliary pools of ADCY5 and ADCY6 are reduced in *Ankmy2* knockout kidney epithelial cells. (A-D)** ADCY5^LAP and ^LAPADCY6 were localized to cilia in stably expressing IMCD3 FlpIn cell line and lacked them in cilia upon CRISPR-based *Ankmy2* ko, as detected upon performing immunostaining with antibodies against GFP and acetylated tubulin. Scale: 5 μm. Quantification in **(B, D)** shown as mean±SD. Total 246-2000 cells counted/cell line and comparison was done by one-way ANOVA with Sidak's multiple comparisons tests. Arrows and arrowheads show GFP positive and GFP negative cilia, respectively. ****, P<0.0001 with respect to control IMCD3; ns, non-significant. **(E-F)** Immunoblots showing glycosylation state of stably expressed LAP-tagged Adenylyl cyclases in IMCD3 FlpIn cells after tandem affinity purification followed by Endo H/PNGase treatment (flow-chart in **E**). Form "c", complex glycosylated; Form "b", core glycosylated; Form "a", non-glycosylated. Abbreviations: DDM, n-Dodecyl-β-D-Maltoside; CHS, Cholesteryl hemisuccinate. **(G)** Immunoblots showing glycosylated states of stably expressed LAP-tagged AC5 and AC6 present in control and *Ankmy2* ko±^HAANKMY2 IMCD3 FlpIn cells as shown in **(E)**. Immunoblots for ANKMY2 and ^HAANKMY2 in lysates along with α-tubulin are shown below. Data showing ratios of complex glycosylated immunoprecipitated form vs tubulin in inputs from N=2-4 experiments, quantified as mean±SD. See also S5 Fig.

Finally, we tested normal human kidney (NHK) and human ADPKD cyst-lining primary cells (Methods). Previous studies have suggested a role of ADCY3 in determining cAMP levels in human ADPKD cells [51]. We detected endogenous ADCY3 localization in both NHK and ADPKD cilia and further evaluated whether ciliary localization depended on ANKMY2 using RNAi. We found that efficient knockdown of *ANKMY2* significantly reduced ciliary levels of ADCY3 in both NHK and ADPKD cells compared to control siRNA-treated cells (S5D-S5F Fig).

Overall, ANKMY2 directs ciliary localization of ADCY5, ADCY6 and ADCY3 in renal epithelial cells and is a core biological mechanism in these cells (as we showed previously for neuroepithelial cells [56]). Finally, ANKMY2 operates the same way in healthy cells as it does in diseased cells from PKD patients.

## Ciliary elongation induced by PC1 loss in cystic kidney epithelia is suppressed by ANKMY2 loss

PC1/2 loss increases cilia length in both ADPKD mouse models [13,36–38] and human patients [38]. Therefore, we tested how ciliary length changes in the established cysts and whether ANKMY2 continues to mediate such regulation, particularly with relation to the sex-specific dependance of ANKMY2 on adult-onset cystic phenotype. We co-immunostained kidney sections with cilia marker acetylated tubulin in epithelial LTL- or AQP2-stained cells to quantify ciliary length in proximal tubule and collecting duct epithelia, respectively (Fig 6A–6C) at 5 months following Doxycycline injections at P28-30. For this quantification, we chose kidneys from male and female mice of respective genotypes with cystic indices across the spectrum (Fig 6D). Overall, we found that cilia were shorter in the proximal tubules compared to the collecting ducts, and with respect to corresponding tubule segments in 10 week old mice, as reported before [67] (Fig 6B–6C). The kidney epithelia of *Pkd1 cko* male mice showed significantly elongated cilia compared to controls in both proximal tubule (by ~3.1 µm) and collecting duct regions (by ~2.5 µm) compared to controls (Fig 6B–6C). Such increase was also apparent in female animals (Fig 6C). The ciliary lengthening correlated with increasing cystic index, scaling with disease severity in both *Pkd1 cko* male and female mice (Fig 6D). Interestingly, *Dko* mice showed significantly reduced ciliary length compared to *Pkd1 cko* male and female mice in both tubule segments (Fig 6B–6C). Further, cilia length in the *Dko* male and female animals remained stable and unaltered in severely cystic kidneys (Fig 6D). In contrast, total cAMP levels increased with cystic burden in both sexes and such increase was apparent in both *Pkd1 cko* and *Dko* animals (Fig 6E), indicating that global cAMP accumulation occurs independently of ANKMY2 and ciliary elongation. These findings reveal that pronounced ciliary lengthening occurs during cyst progression in *Pkd1 cko* mice that accompanies increased cAMP levels, suggesting role of cilia-diffusible cellular cAMP levels in mediating ciliary length increase. However, the absence of ciliary elongation in *Dko* mice, irrespective of sex and high cyst burden, demonstrates that increased cellular cAMP levels alone are insufficient to induce ciliary lengthening in the absence of ANKMY2.

## Ciliary length increase from PC1 loss in kidney epithelia precedes overt cystogenesis

Optogenetic activation of a cilium targeted bacterial photoactivable adenylyl cyclase after activation elevates ciliary cAMP levels, increased ciliary length and cystic changes in 3D cultured cells [43,44], suggesting the role of ciliary length regulation in initiation of cystogenesis. However, the relevance of ciliary length control in cystogenesis *in vivo* is not clear, and the causal relationship between ciliary elongation and cyst development has yet to be defined. As ANKMY2 regulates ciliary localization of adenylyl cyclases and increased total cAMP levels alone were insufficient to induce ciliary lengthening in cysts in the absence of ANKMY2, we examined time-resolved cilia length changes in the adult-onset PKD models prior to overt cystic changes and total cAMP increases. We reasoned that ANKMY2 dependence on changes in cilia length would highlight involvement of ciliary adenylyl cyclase signaling preceding and/or accompanying cystogenesis. Cilia length changes in *Pkd1 cko* mice preceding cyst progression would argue for ciliary involvement preceding overt cystogenesis. Therefore, we checked kidneys at 8–14 weeks following Doxycycline injections at P28-30 and measured cilia lengths in proximal tubules and collecting ducts in both sexes (Fig 7A). We co-immunostained kidney sections with cilia marker acetylated tubulin in epithelial LTL- or AQP2-stained cells to quantify ciliary length in proximal tubule and collecting duct epithelia, respectively.

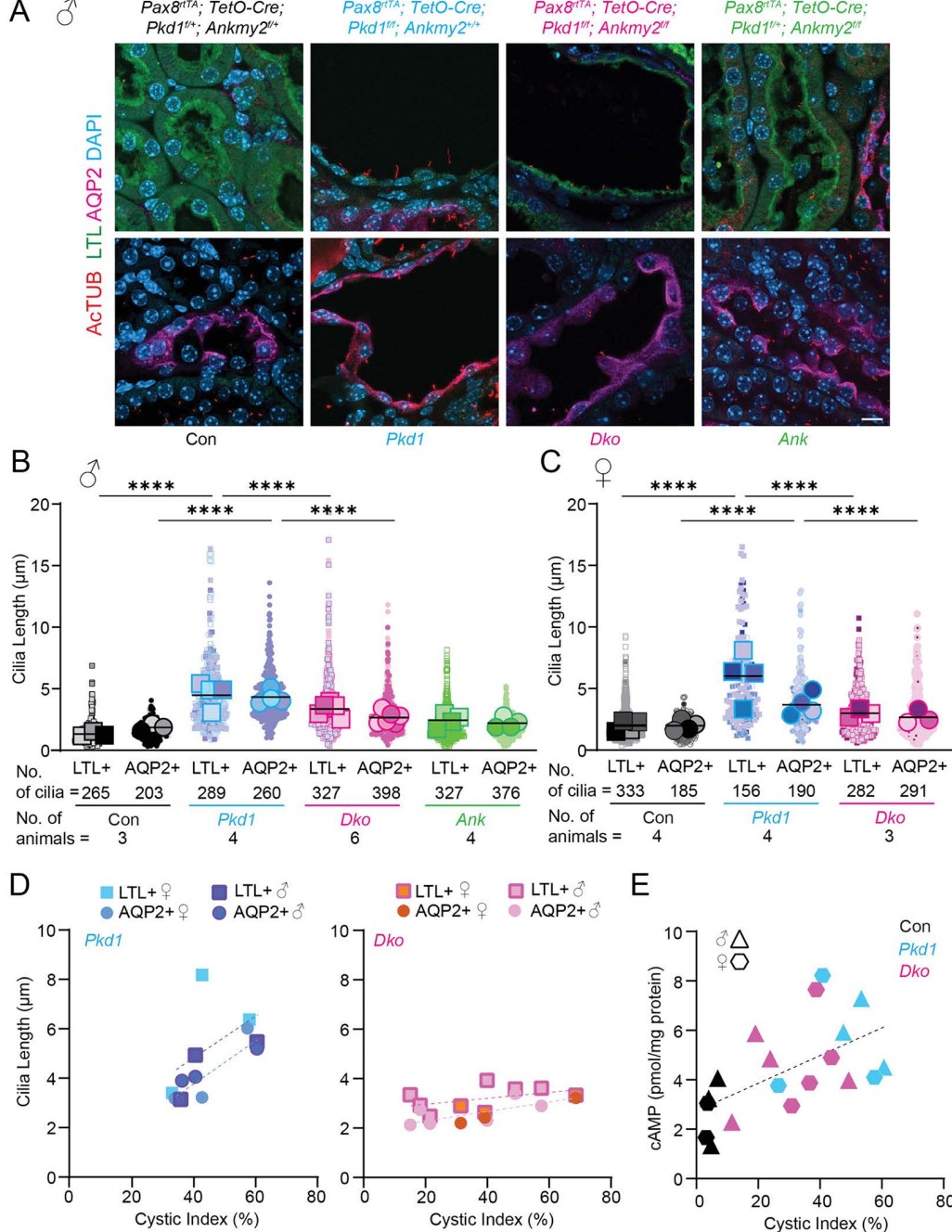

**Fig 6. Ciliary length increase from PC1 loss in adult kidney cyst epithelia is suppressed from ANKMY2 loss. (A)** Kidney sections of 5-month-old mice (as in Fig 3A) were immunostained for acetylated tubulin, AQP2 and LTL and counterstained with DAPI. Scheme for inducible conditional knockout in kidney nephron epithelia in the adult-onset model was like Fig 3A. Scale 10 μm. **(B-C)** Cilia lengths were quantified and super plots of N = 3-6 male (B) or N = 3-4 female (C) 5-month-old mice shown for each genotype. Individual cilia lengths from each kidney are shown with identical shapes and averages from each kidney are plotted with larger analogous shapes. Horizontal bars, mean of average cilia lengths from each kidney. ****, P < 0.0001 using one-way ANOVA with Sidak's multiple comparisons tests for comparing between all cilia. **(D)** Mean cilia lengths with respect to cystic indices shown for 5-month-old mice. Cilia length in $Pax8^{rtTA}$; $TetO$-$Cre$; $Pkd1^{f/f}$ animals scale with cystic indices but remains unchanged irrespective of cystic indices in $Pax8^{rtTA}$; $TetO$-$Cre$; $Pkd1^{f/f}$; $Ankmy2^{f/f}$. Both males and females are shown. **(E)** Total cAMP levels with respect to cystic indices shown for respective genotypes for 5-month-old mice and the levels correlated with cystic index across genotypes irrespective of sex.

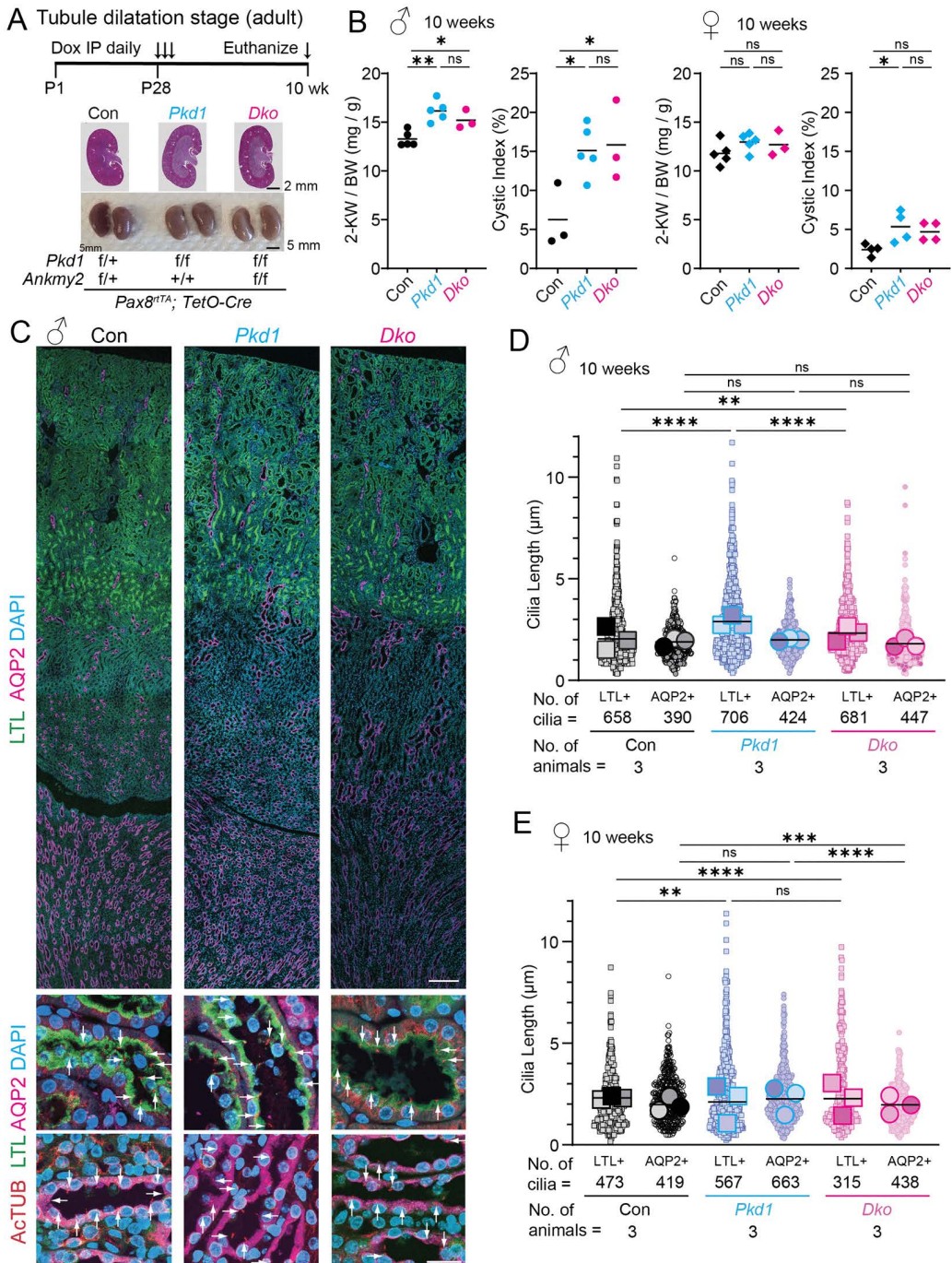

**Fig 7. Ciliary length increase from PC1 loss in adult kidney epithelia precedes overt cystogenesis and is suppressed from ANKMY2 loss. (A)** Scheme for inducible conditional knockout in kidney nephron epithelia in an adult-onset model of PKD to detect "pre-cystic" tubules. H&E and whole mount images of 10-week-old kidneys in control, *Pax8^rtTA; TetO-Cre; Pkd1^f/f* and *Pax8^rtTA; TetO-Cre; Pkd1^f/f; Ankmy2^f/f* male mice in C57BL/6J background. **(B)** 2-kidney to body weight ratios, and cystic indices of 8 to 10-week-old kidneys in control, *Pax8^rtTA; TetO-Cre; Pkd1^f/f* and *Pax8^rtTA; TetO-Cre; Pkd1^f/f; Ankmy2^f/f* male and female mice in C57BL/6J background shown. Horizontal bars, mean. **, P < 0.01; *, P < 0.05 using one-way ANOVA with Sidak's multiple comparisons tests. **(C)** Kidney sections were immunostained for acetylated tubulin, AQP2 and LTL and counterstained with DAPI in 10-week-old male mice. Magnified images of cortical and cortico-medullar junctions are shown below. Scale: Top whole kidney panels, 200 μm; Lower panels, 20 μm. **(D-E)** Cilia lengths were quantified and super plots of N = 3 male **(D)** or female **(E)** 10-week-old animals shown for each genotype. Individual cilia lengths from each kidney are shown with identical shapes and averages from each kidney are plotted with larger analogous shapes. Horizontal bars, mean of

averages from each kidney. ****, P<0.0001; ***, P<0.001; **, P<0.01; ns, not significant using one-way ANOVA with Sidak's multiple comparisons tests for comparing between all cilia. See also S6 and S7 Figs.

By 10 weeks both the 2-kidney/body weight ratios and kidney cystic indices showed significant but only modest differences between control and *Pkd1 cko* males (Figs 7B **and** S6). The kidney tubules in *Pkd1 cko* mice just started to show dilatations as reported earlier [68], but overt cysts were not seen (Figs 7A–7C, S6**, and** S7), a condition we call "tubule dilatation". However, by 10 weeks we found that the cilia lengths in the proximal tubule epithelial cells of *Pkd1 cko* males were significantly increased by ~0.9 µm than control males (Figs 7C–7D**, and** S6). Cilia lengths were also significantly increased in *Pkd1 cko* males than *Dko* males, despite no differences in cystic index at this stage (Fig 7B–7D). Although we did not detect changes in cilia lengths in collecting duct epithelial cilia at 10 weeks (Fig 7C–7D), by 14 weeks significant increase in cilia lengths was apparent in the collecting duct epithelial cilia of *Pkd1 cko* male kidneys (compared to control kidneys (S6 Fig). These results suggest segment-specific dynamics in cilia remodeling during initial cystogenesis.

Female mice developed cysts slowly, as reported [69,70], and at 10–14 weeks, the 2-kidney/body weight ratios were not significantly different between control, *Pkd1 cko* and *Dko* females, although kidney cystic indices showed significant but only modest differences between control and *Pkd1 cko* (Figs 7B, S6**, and** S7). Although female animals showed significantly different cilia lengths in the proximal tubule epithelial cells between controls and *Pkd1 cko*, the average cilia length between these genotypes at this stage remained similar (Fig 7E), suggesting slower ciliary lengthening from the delayed cyst initiation dynamics. By 14 weeks, significant increase in the proximal duct epithelial cilia of *Pkd1 cko* female kidneys was apparent compared to control kidneys (S6 Fig).

The initiation of cilia length increases in proximal tubule and collecting duct epithelia of *Pkd1 cko* mice demonstrate ciliary length changes preceding overt cystogenesis and occurs during the tubule dilatation phase in an ANKMY2-dependent manner. Given that *Dko* mice did not show cilia elongation irrespective of cyst/ total cAMP status (Fig 6), we conclude ciliary elongation during cyst progression is not a passive consequence of elevated cAMP or cyst growth but is contingent upon ANKMY2-dependent initial ciliary lengthening in the tubule dilatation phase. Overall, these findings support a role for ciliary length control during tubule dilatation and identify ANKMY2 as a critical regulator of early disease events in ADPKD.

## Discussion

The ciliary components that are derepressed upon loss of polycystins in cystogenesis have remained elusive. By definition, a cilia dependent cyst activator should be effective in diverse ADPKD models with varying developmental onset. Our results demonstrate that lack of ANKMY2 suppresses cystogenesis in both embryonic-onset or adult-onset ADPKD mice models. Loss of ANKMY2 prevents early embryonic onset cystogenesis and extends lifespan. Cyst reduction in adult-onset ADPKD males was also seen in both proximal tubules and collecting ducts, indicating a broad nephron-wide effect. In contrast, disruption of other ciliary trafficking regulators—such as TULP3 or ARL13B, which are trafficking nodes for multiple ciliary cargoes [11,71], yield variable effects in different ADPKD models [12,14,16,72]. Whereas *Tulp3* deletion reduces cystogenesis in adult-onset ADPKD models [12], embryonic loss of *Tulp3* exacerbates cystic burden from concomitant deletion of *Pkd1* [12,14]. Part of this exacerbation is likely from lack of TULP3 cargo ARL13B from cilia, based on the dynamics of ARL13B loss from cilia in early postnatal cysts [14,71] and cystic phenotype in mice lacking ciliary pools of ARL13B [16]. Although loss of ANKMY2 does not fully reverse final cyst progression in embryonic-onset ADPKD and only partially ameliorates disease in adult-onset disease, our results strongly suggest that adenylyl cyclases are key ciliary factors—though not the sole contributors—that are derepressed upon polycystin loss in both embryonic- and adult-onset disease.

Mechanistically, lack of ANKMY2 prevented localization of multiple adenylyl cyclases to cilia and plasma membrane of mouse and human renal epithelial cells, while drastically reducing–but still retaining–mature glycoslylated forms. While

the effects on ANKMY2 loss on mature glycoslylated levels of adenylyl cyclases contributes to lack in plasma membrane localization, such reduction was not sufficient to circumvent later changes in adenylyl cyclase/cAMP signaling in cystic kidneys from *Pkd1* loss. For example, pCREB levels in collecting duct epithelia in later postnatal kidneys at P15 were similar between *Hoxb7-Cre*; *Pkd1^flf^* and *Hoxb7-Cre*; *Pkd1^flf^*; *Ankmy2^flf^* animals. Similarly, cystogenesis persisted in adult-onset ADPKD models in female mice despite ANKMY2 loss and was accompanied by high cAMP levels in kidney. Given the effect on ciliary length control in adult-onset ADPKD in an ANKMY2-dependent manner (see next section), and ciliary length control being mediated by factors that localize in the cilia-centrosomal complex [34,35], the most parsimonious model would be that the effect on ciliary/peri-ciliary adenylyl cyclase signaling by ANKMY2 is the major contributing factor in regulating cystic burden in ADPKD (Fig 8). Similarly, high hedgehog signaling from pathway derepression in the *Ankmy2* ko mouse neural tube was fully cilia-dependent [56]. In the kidney tubules, lack of ciliary/peri-ciliary adenylyl cyclase signaling from ANKMY2 loss likely suppresses cyst initiation but could be less effective at later stages when cellular cAMP levels rise during cyst progression. Finally, as ANKMY2 operates the same way in healthy human renal epithelial cells as it does in diseased cyst-lining cells from PKD patients, the ANKMY2-dependent trafficking pathway is not created during ADPKD pathogenesis. Instead, loss of PC1 likely hijacks this pre-existing, normal signaling compartmentalization mechanism to initiate cysts. Overall, these data strengthen the central hypothesis that ciliary AC localization is a key permissive event during cyst initiation in ADPKD.

## Cilia length control suggests distinct cyst initiation and progression mechanisms

One critical index of ciliary regulation during adult-onset ADPKD initiation and progression involves ciliary length control by dysregulated polycystin signaling in an ANKMY2-dependent manner. The following data highlight the importance of ciliary length control in cyst pathogenesis. First, ciliary elongation initiated at the tubule dilatation stage in both proximal tubule and collecting duct epithelia of *Pax8^rtTA^*; *TetO-Cre*; *Pkd1^flf^* animals by 10–14 weeks in males. This early lengthening precedes overt cystogenesis and is absent upon loss of *Ankmy2*, indicating an ANKMY2-dependent process. At this early stage, total cAMP levels are not elevated, so this initial ciliary lengthening is likely from ciliary adenylyl cyclases and coincidental lack of polycystin signaling. Second, during advanced cystic stages in adult ADPKD models, cilia undergo more dramatic elongation—up to ~2.5–3.1 μm, depending on the nephron segment. This progressive lengthening scaled with cystic index and total cAMP levels in *Pax8^rtTA^*; *TetO-Cre*; *Pkd1^flf^* animals suggesting diffusion of cytoplasmic cAMP from cystic epithelia into cilia in the continued ciliary length increase. Third, cilia length remained unchanged in *Pax8^rtTA^*; *TetO-Cre*; *Pkd1^flf^*; *Ankmy2^flf^* mice despite high cystic load and kidney cAMP levels suggesting that cytoplasmic cAMP alone is insufficient to trigger ciliary elongation in absence of ANKMY2. Instead, early permissive changes—specifically ANKMY2-dependent ciliary lengthening during the tubule dilatation stage—enable the dramatic cilia elongation observed in advanced cysts, highlighting the likely functional specificity of cilia compartmentalized adenylyl cyclase signaling in disease initiation.

As loss of ANKMY2 results in reduced ADCY3/5/6 levels in renal epithelial cilia, taken together, our results implicate ANKMY2-mediated trafficking of adenylyl cyclases to the cilium as essential for initiating ciliary lengthening following *Pkd1* deletion. We propose a two-phase model: (1) an early, cilia-autonomous phase in which localized adenylyl cyclase signaling and coincidental lack of polycystin signaling—potentially acting through unidentified ciliary effector(s)—drives initial ciliary lengthening; followed by (2) a later phase in which rising cytoplasmic cAMP in cystic epithelia diffuses into or otherwise influences the ciliary compartment, further amplifying cilia length (Fig 8). Overall, these findings support a dual mechanism of adenylyl cyclase regulation in ADPKD: ANKMY2-mediated cilia-dependent signaling initiates cystogenic changes, while global cAMP accumulation fuels later progression.

Based on our *in vivo* findings, we propose criteria for ciliary effector(s) driving cyst initiation and ciliary length controller(s) during cyst progression (Fig 8). A ciliary effector driving cyst initiation should be a ciliary/peri-centrosomal localized factor, a cilia length regulator and regulatable at cilia by adenylyl cyclase signaling. Critically, such an effector should

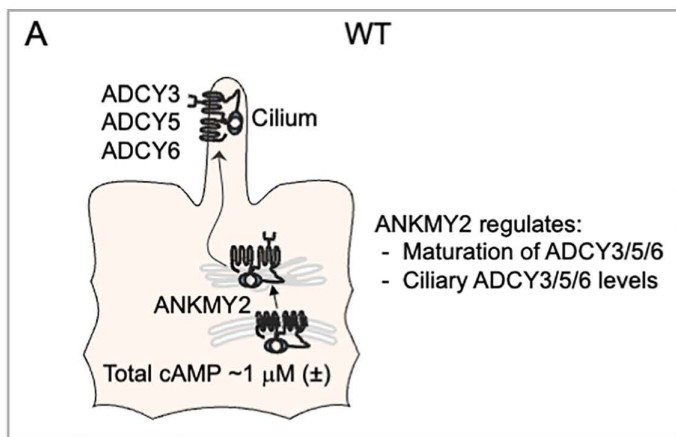

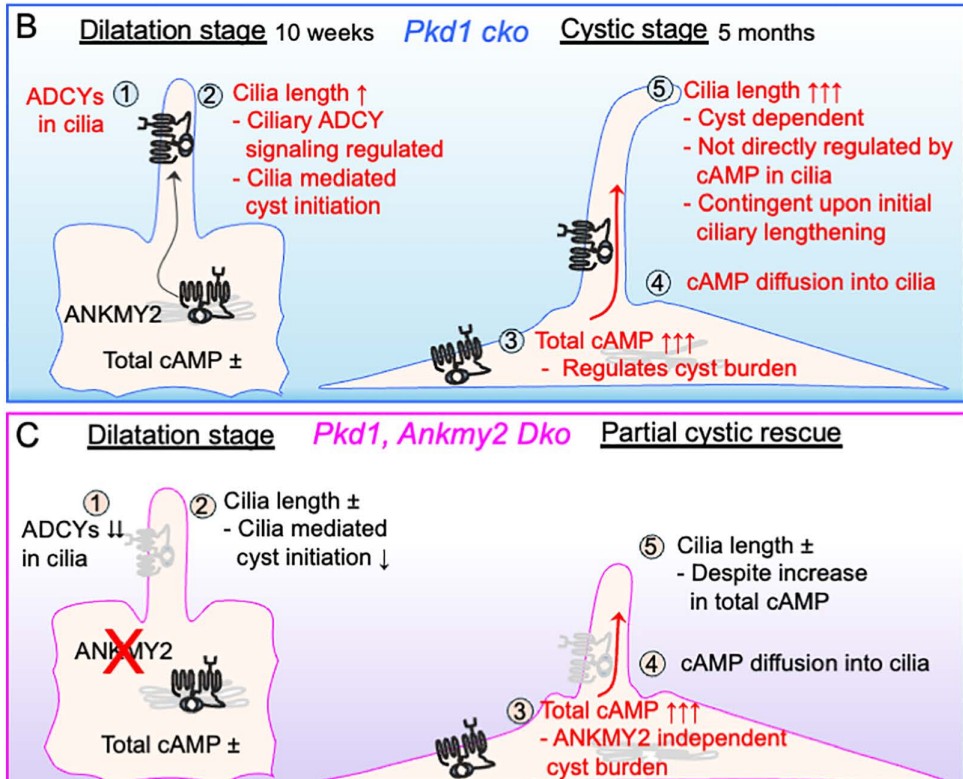

**Fig 8. Model showing de-repression of an ANKMY2-dependent ciliary adenylyl cyclase signaling pathway preceding cystogenesis in adult-onset PKD.** In wild-type, ANKMY2 functions in maturation and localization of adenylyl cyclases ADCY3/ADCY5/ADCY6 to kidney epithelial cilia **(A).** At the tubule dilatation stage in adult-onset *Pax8^{rtTA}; TetO-Cre; Pkd1^{f/f}* (*Pkd1 cko*) kidneys **(B)**, we note initiation of ciliary lengthening, suggesting that ciliary changes precede cyst formation. Cilia lengths are drastically increased in cystic adult-onset *Pkd1* kidneys. In adult-onset *Pax8^{rtTA}; TetO-Cre; Pkd1^{f/f}; Ankmy2^{f/f}* (*Dko*) kidneys the initial cilia lengths are not increased **(C)**. These results suggest de-repression of an ANKMY2-dependent ciliary adenylyl cyclase signaling pathway preceding cystogenesis in *Pkd1* mice (Steps 1, 2) and high ciliary cAMP diffusing from cytoplasm at later cystic stages (Step 3, 4) contributing to further ciliary lengthening (Step 5). Cilia lengths are not increased in the adult-onset *Dko* mice with bigger cysts having high cellular cAMP levels. Thus, permissive changes from ciliary adenylyl cyclase targeting by ANKMY2 during the tubule dilatation stage (Step 2) ultimately dictate the massive ciliary lengthening in adult-onset *Pkd1 cko* mice (Step 5). A ciliary effector in cyst initiation (Step 2) should be a cilia-localized cAMP-regulatable cilia length regulator, the ciliary localization of which should increase prior to overt cystogenesis. While plasma membrane localization of ADCYs are affected in *Ankmy2* ko cells, the presence of cyst initiating ciliary effector(s) mediating cilia elongation during dilatation stage argues for a predominant role for ciliary/periciliary adenylyl cyclase signaling by ANKMY2, as in the neural tube [56]. However, while a ciliary length controller during cyst progression (Step 5) should be localized to cilia in cysts, it might not require to be strictly cAMP regulatable at cilia. Total cAMP increase in cysts is ANKMY2 independent and regulates cystic processes such as fluid secretion by CFTR [25–27] and cell proliferation through B-Raf/MEK/ERK pathway [28]. The

slower initial ciliary lengthening in female adult-onset *Pkd1 cko* mice suggest that the ciliary component of cAMP signaling derepressed from polycystin loss is temporally delayed and/or overlaps with downstream cytoplasmic pathways precluding efficient ANKMY2-mediated cyst suppression.

exhibit dynamic regulation at the tubule dilatation stage—prior to overt cystogenesis—positioning it as an early driver of disease. Conversely, a ciliary length controller during cyst progression may need not be directly cAMP regulated within this compartment.

Loss of ANKMY2 suppresses cystogenesis in adult-onset ADPKD model in male mice, but not in females. The sex-specificity aligns with the well-documented sex-related differences in both human ADPKD patients and rodent models, which exhibit more severe cystic disease in males than in females [69,70]. Such sex-specificity in renal disease progression is secondary to differences in the levels of gonadal hormones that control androgen receptor mediated control of sexually dimorphic gene expression in tubule segments [60,61]. In *Pax8^rtTA*; *TetO-Cre*; *Pkd1^f/f* females, ciliary lengthening in the proximal tubule at the tubule dilatation stage was less robust compared to males. The slower initial ciliary lengthening during the tubule dilatation stages in females suggest that the ciliary component of adenylyl cyclase signaling derepressed from polycystin loss is temporally delayed and/or overlaps with downstream cytoplasmic pathways precluding efficient ANKMY2-mediated suppression of cystic burden.

The extent of ciliary length increase can also potentially be a marker for impending cyst formation and cyst severity in ADPKD [38]. Loss of PC1/2 has been shown to increase cilia length in ADPKD mouse models [13,36,37] and in human patients [38]. Importantly, optogenetic activation of a cilium targeted bacterial photoactivable adenylyl cyclase elevates ciliary cAMP levels and induces ciliary lengthening within one hour [43,44]. Similarly, pharmacological elevation of cellular cAMP using the adenylyl cyclase activator Forskolin for a few hours increases ciliary length in cultured mammalian kidney epithelial cells [42]. However, this effect is context-dependent: under conditions mimicking physiological tubular flow [67] forskolin-induced ciliary elongation is blocked [42], suggesting that normal flow-mediated signaling actively restrains ciliary length in a healthy nephron. Moreover, ciliary disassembly processes could regulate ciliary signaling and length [73]. For example, loss of polycystins suppresses deciliation via the activation of the centrosomal integrity pathway [74], potentially stabilizing elongated cilia and promoting sustained signaling.

Together, these findings suggest a likely pathological feedback loop in ADPKD: *Pkd1* deletion leads to unchecked ciliary adenylyl cyclase signaling and elongation, impairing flow-sensing and disrupting normal flow-dependent ciliary shortening. Failure in feedback control could lock cilia in an elongated, hyper-signaling state—reinforcing cyst promoting signals in a self-amplifying cycle. Finally, as *ANKMY2* knockdown similarly reduces ciliary ADCY3 levels in both normal human kidney cells and human ADPKD cyst-lining cells, targeting ANKMY2 can be highly significant for translational relevance. Our data confirms that the ANKMY2 and the adenylyl cyclase trafficking pathway is conserved in humans, making ANKMY2 a more compelling therapeutic target in ADPKD in reducing cystic burden.

## Limitations of the current study

Currently, we are limited by antibodies that are unable to detect ciliary levels of ACs in kidney epithelia. Therefore, we are unable to extrapolate our cell culture studies using kidney epithelial cell lines into *in vivo* contexts of kidney homeostasis and cystogenesis. Given that PC1-PC2 functions in WNT mediated $Ca^{2+}$ entry [75] and AC5/6 is inhibited by $Ca^{2+}$ and $G\beta\gamma$ subunits downstream of polycystins [76–78], PC1/2 activation is expected to reduce cAMP levels in cilia. However, quantifying cAMP levels in cilium will require elucidation of mechanisms underlying activation of PC1-PC2 in cilia. Measuring *basal* steady-state cAMP levels in cilia might not resolve temporal cAMP changes from either PC1/2 or ANKMY2 loss. Our study also highlights tubule segment-specific mechanisms in dynamics of cilia length regulation during the pre-cystic stage. Elongated cilia could reflect signaling by ciliary and periciliary effectors promoting cystogenesis. Alternatively, elongated cilia could alter cilia-generated signaling by modulating the activity of ciliary and periciliary effectors induced by flow.

These effectors could potentially influence early transcriptional and post-transcriptional programs promoting cystogenesis [19,79,80]. In the hedgehog pathway, ciliary and cytoplasmic cAMP levels are differentially interpreted in vertebrate cells in regulating signaling [31] and tissue-specific phenotypes [81,82]. In fact, post-transcriptional regulation of targets by mechanical transduction in crown cilia in the embryonic node is also pivotal in left right asymmetry [83,84]. However, the specific ciliary adenylyl cyclase signaling regulated targets in ADPKD cystogenesis are currently unknown and represent a key area for future studies. Overall, our work provides a framework for identifying and testing adenylyl cyclase regulated ciliary effectors and early subcellular events that initiate cystogenesis.

## Materials and methods

### Ethics statement

All the animals in the study were handled according to protocols approved by the UT Southwestern Institutional Animal Care and Use Committee (APN 2016–101516), and the mouse colonies were maintained in a barrier facility at UT Southwestern, in agreement with the State of Texas legal and ethical standards of animal care.

### Mouse strains and genotyping

All mice were housed at the Animal Resource Center of the University of Texas Southwestern (UTSW) Medical Center. Mice were housed in standard cages that contained three to five mice per cage, with water and standard diet *ad libitum* and a 12 h light/dark cycle. Both male and female mice were analyzed in all experiments. *HoxB7-Cre* mice were obtained from O'Brien Kidney Research Core of UT Southwestern. *Pkd1^f/f* allele has been described before [62] and were backcrossed to C57BL/6J background. *Pax8^rtTA*; *TetO-Cre* [10] combined with *Pkd1^f/f* allele [85] that were back crossed to C57BL/6J were obtained from PKD RRC, University of Maryland, School of Medicine. Mice were housed in standard cages that contained three to five mice per cage, with water and standard diet *ad libitum* and a 12 h light/dark cycle.

### Mouse ES cells

ES cells targeting *Ankmy2* (NM_146033.3, MGI: 2144755; third exon flanked by LoxP sites) that was generated by homologous recombination in mouse ES cells of the C57BL/6 strain were from EUCOMM (Clone # HEPD0679–6-C03) [56]. ES cells were grown on SNL feeders with media containing 20% Serum, 6 mM L-glutamine, 1X Penicillin/Streptomycin, 1 mM β-mercaptoethanol, 1 mM Non-essential Amino Acids, 1X Nucleosides, 10 mg/L Sodium Pyruvate, ESGRO supplement 66 μl/L and incubated at 37 °C in 5% $CO_2$ (Dr. Robert Hammer lab, UT Southwestern Medical Center, Dallas). The ES cells were injected into host embryos of the C57BL/6 albino strain by the transgenic core (Dr. Robert Hammer lab, UT Southwestern Medical Center, Dallas.

### Mouse genotyping

To genotype *Ankmy2* mice, following primers were used. For floxed allele with or without deletion: 3F (5'-CTG TCT CCA TAT TCA CAC ATT GAA TAG C-3'), 4R (5'-GCT GCA TGC ATC AAA GGA GTC ATT CC-3') and 2R (5'-TGA ACT GAT GGC GAG CTC AGA CC-3') gave 508 bp for wild type, 732 bp for floxed allele, and 289 bp for deleted floxed allele (cko). *Cre* allele was genotyped with Cre-F (5'-AAT GCT GTC ACT TGG TCG TGG C-3') and Cre-R (5'-GAA AAT GCT TCT GTC CGT TTG C-3') primers (100 bp amplicon). For the *Pkd1 ^tm2Som* [62] allele, forward (5'-CCG CTG TGT CTC AGT GTC TG-3') and reverse (5'-CAA GAG GGC TTT TCT TGC TG-3') gave 400 bp for floxed allele and 200 bp for wild type. For the *Pkd1^tm2Ggg* allele [85], F4 forward (5'-CCT GCC TTG CTC TAC TTT CC-3') and R5 back (5'-AGG GCT TTT CTT GCT GGT CT-3') gave 250 bp for floxed allele and 180 bp for wild type. For *Pax8^rtTA*, oIMR7385 (5'-CCA TGT CTA GAC TGG ACA AGA-3') and oIMR7386 (5'-CTC CAG GCC ACA TAT GAT TAG-3') gave 595 bp for *Pax8^rtTA* allele.

## Tissue processing, antibodies, immunostaining and microscopy

Mice were perfused with PBS, and the kidneys were dissected and fixed in 4% paraformaldehyde overnight at 4°C and processed for cryosection or paraffin embedding and sectioning. For cryosection, the kidneys were incubated in 30% sucrose for 1–2 days at 4°C. Kidneys were mounted with OCT compound and cut into 15 µm frozen sections. For paraffin section, kidneys were processed over a 12-hour period using a Thermo-Fisher Excelsior Automated Tissue Processor (A82300001; ThermoFisher Scientific), which dehydrated the kidneys through 6 ethanol concentrations, from 50% ethanol to 100% ethanol, cleared through 3 changes of xylene, and infiltrated with wax through 3 Paraplast Plus paraffin baths (39602004; Leica). Samples were embedded in Paraplast Plus using paraffin-filled stainless steel base molds and a Thermo-Shandon Histocenter 2 Embedding Workstation (6400012D; ThermoFisher Scientific). The kidneys were then cut in 5 µm thick sections, deparaffined and treated with microwave in Antigen Retrieval citra solution (HK086-9K; BioGenex. Fremont, CA) for 10 min. For frozen sections, the sections were incubated in PBS for 15 min to dissolve away the OCT. Sections were then blocked in blocking buffer (1% normal donkey serum [Jackson immunoResearch, West Grove, PA], in PBS) for 1 hour at room temperature. Sections were incubated with primary antibodies against the following antigens; overnight at room temperature or 4C: Acetylated tubulin (T6793; Sigma mouse IgG2b, 1:500), AQP2 (SC515770, Santa Cruz Biotechnology, mouse IgG1, 1:500), Ki67 (ab16667, Abcam, 1:500), pCREB (9198S; Cell Signaling, 1:500). After three PBS washes, the sections were incubated in secondary antibodies (Alexa Fluor 488-, 555-, 594-, 647- conjugated secondary antibodies, or anti-mouse IgG isotype-specific secondary antibodies; 1:500; Life Technologies, Carlsbad, CA or Jackson ImmunoResearch) or cell surface markers, Fluorescein labeled Lotus tetragolonobus lectin (LTL; 1:200, FL 1321–2 Vector laboratories) for 1 hour at room temperature. Cell nuclei were stained with DAPI (Sigma) or Hoechst 33342 (Life technologies). Slides were mounted with Fluoromount-G (0100–01; Southern Biotech) and images were acquired with a Zeiss AxioImager.Z1 microscope or a Zeiss LSM980 confocal microscope. For hematoxylin and eosin staining, paraffin sections were stained by hematoxylin and eosin (Hematoxylin 560; 3801575; Leica and Alcoholic Eosin Y 515; 3801615; Leica) using a Sakura DRS-601 x-y-z robotic-stainer (DRS-601; Sakura-FineTek, Torrance, CA). Slides were dehydrated and mounted with Permount (SP15–100; ThermoFisher Scientific). For immunofluorescence experiments in cell lines, cells were cultured on coverslips until confluent and starved for indicated periods before fixation. Cells were fixed with 4% PFA. After blocking with 5% normal donkey serum, the cells were incubated with primary antibody solutions for 1 h at room temperature followed by treatment with secondary antibodies for 30 min along with DAPI. Primary antibodies used were against the following antigens: GFP (Abcam ab13970; 1:10,000), Acetylated tubulin (T6793 Sigma mouse IgG2b, 1:2000), γ-tubulin (sc-17787; Santacruz Biotech mouse IgG2a, 1:500), β-catenin (610154; BD Biosciences, 1:500), HA (clone 3F10; Roche, 1:500). Coverslips were mounted with Fluoromount-G and images were acquired with a Zeiss AxioImager.Z1 microscope. Immunostaining of human primary cells are described in a later section on human primary cells.

## Cell culture and generation of stable cell lines

IMCD3 Flp-In, Phoenix A (PhA, Indiana University National Gene Vector Biorepository), and 293FT cells were cultured in DMEM high glucose (Sigma-Aldrich; supplemented with 10% cosmic serum, 0.05 mg/ml penicillin, 0.05 mg/ml streptomycin, and 4.5 mM glutamine). They have tested negative for Mycoplasma using the Mycoplasma PCR Detection Kit (Genlantis). Transfection of plasmids was done with Polyfect (QIAGEN) or Polyethylenimine (PEI) max. Stable cell lines were generated by retroviral infection or transfection. In many cases, stable lines were flow sorted and further selected for GFP. IMCD3 FlpIn cells stably expressing LAP tag or LAP tagged Adenylyl cyclases were generated by retroviral infection, antibiotic selection, and flow sorting. CRISPR/Cas9 knockout or knockdown lines for *Ankmy2* were generated in IMCD3 Flp-In cells by using guide RNA targeting sequences AAGGAACTGCTGGAAGTGAT (Exon 1) or GAATGTTCATGTCAACTGCT (Exon 2). Clonal lines were tested for knockout or knockdown by Sanger sequencing and immunoblotting for Ankmy2. ORF clones were as follows: ADCY5 (gift of Ron Taussig, UT Southwestern), ADCY6 (Life Technologies IOH40476),

ANKMY2 (Origene RG206770; NM_020319 in PCMV6-AC-GFP vector from Origene). Gateway pENTR clones were generated by PCR cloning and BP reaction as necessary for N- or C-terminal tagging. Gatewaytized pBABE-LAP-N terminus and pBABE-LAP-C terminus plasmids were generated from LAP1 and LAP5 vectors (Addgene) and pBABE puro. We cloned 3×HA-ANKMY2 into pQXIN (Clontech), which was used for retroviral infection in knockout lines expressing LAP-tagged Adenylyl cyclases. Antibiotic selection was used to generate rescue lines stably expressing 3×HA-ANKMY2.

## Cystic index quantification

Cystic index was quantified in paraffin embedded mid-sagittal sections of whole kidneys. HE stained kidney sections were photographed by the PrimeHisto XE slide scanner (Pacific Imaging, Inc.) using HistoView Software. ImageJ software (National Institutes of Health, Bethesda, MD) was used to calculate the cystic index. The images were converted to 8-bit grey scale. Equal sized non-overlapping areas were cropped covering the entire kidney image. The image threshold was adjusted similarly, and the percentage of black area was analyzed in each cropped image. Data from all cropped areas from the kidney were averaged and finally subtracted from 100 to give the cystic index as a percent.

## Tandem affinity purification and immunoblotting

IMCD3 FlpIn cells stably expressing LAP tag or LAP tagged Adenylyl cyclases were lysed in buffer containing 50 mM Tris-HCl, pH 7.4, 200 mM KCl, 1 mM MgCl2, 1mM EGTA, 10% glycerol, 1 mM DTT, 1% digitonin, 0.05% n-Dodecyl-β-D-Maltoside, 0.25% Cholesteryl hemisuccinate, 1 mM of AEBSF, 0.01 mg/mL of Leupeptin, pepstatin and chymostatin [86]. Lysates were centrifuged at 12000xg for 10 min followed by tandem IPs [87]. Briefly, the GFP immunoprecipitates were first digested with TEV (N terminal LAP) or PreScission (for C terminal LAP) protease for 16h at 4°C. The supernatants were subjected to secondary IPs with S-protein agarose. Treatment with Endo H and PNGase F (NEB) was performed on the IP-ed proteins on S-protein agarose beads in NEB designated buffers at 37°C for 2h. The resulting secondary IPs were eluted in 2×urea sample buffer (4 M urea, 4% SDS, 100 mM Tris, pH 6.8, 0.2% bromophenol blue, 20% glycerol, and 200 mM DTT) at 37°C for 30 min and analyzed by immunoblotting [86]. For detection of different glycosylation forms of the LAP-tagged Adenylyl cyclases by S-tag immunoblotting and based on the stable expression levels, we required ~0.75 ml packed cell pellet for finally eluting secondary IPs in 30–40 µl of 2×urea sample buffer from 30-40 µl S-protein agarose beads. Tandem-IPs were mostly run on 4–20% Mini-PROTEAN TGX Precast Protein Gels (Bio-Rad). Immunoblots from tandem affinity purifications were probed with antibodies against S-tag (mouse monoclonal MAC112; EMD Millipore), ANKMY2 (HPA067100, Sigma), α-tubulin (clone DM1A, T6199, Sigma), HA tag (clone 3F10, Roche) followed by visualization using IRDye-tagged secondary antibodies. The images were acquired with the Odyssey Fc Imaging System (LI-COR Biosciences), and the analysis and quantification of individual western blot bands was performed with Image Studio Lite Western Blot Analysis software (LI-COR Biosciences). Immunoblotting from kidney lysates is described in supplemental methods.

## Immunoblotting kidney samples

Frozen whole kidney tissues were homogenized using Fisherbrand Pellet Pestles in a RIPA lysis buffer containing 10 mM sodium phosphate (pH 7.5), 150 mM NaCl, 1.5 mM MgCl2, 0.5 mM DTT, 1% Triton X-100, 10 µg/ml each of Leupeptin, Pepstatin and Chymostatin and 0.1 mM AEBSF), Phosphatase inhibitor cocktail 2 (Sigma-Aldrich, P5726) and Phosphatase inhibitor cocktail 3 (Sigma-Aldrich, P0044). The lysates were centrifuged at 10,000×g for 10 min at 4°C. Protein concentration was determined by bicinchoninic acid assay (Thermo Fisher Scientific). Samples were lysed with SDS buffer containing β-mercaptoethanol, heated at 95°C for 5 min and 30 µg proteins were loaded each wall on Mini-PROTEAN TGX 4–15% gels (BIO-RAD Laboratories, Inc.). Next, proteins were blotted on polyvinylidene difluoride (PVDF) membranes. Membranes were blocked for 1h at room temperature with TBS-Tween (0.1%) containing 1% Gelatin (Sigma), followed by incubation with the primary antibody diluted in TBS-T for overnight at 4°C. Primary antibodies used were mouse

anti-ERK (9107S; Cell Signaling Technology, 1:1,000), rabbit anti-pERK (4370S; Cell Signaling Technology, 1:1,000) and mouse anti α-tubulin (clone DM1A, Sigma; T6199) 1:5,000). Next day, the membranes were washed 5 times with TBS-T and followed by visualization using IRDye-tagged secondary antibodies and hFAB Rhodamine Anti-β-Tubulin (Bio-Rad; 12004166) for counterprobing anti-ERK blots. Images were taken in a BioRad Chemidoc MP imaging system. Densitometry was performed using default settings of the Bio-Rad Image lab software.

## Total cAMP measurements from kidneys

Frozen kidney tissue (~100 mg) was homogenized using 1.4 mm Zirconium-Silicate spheres (Lysing Matrix D beads, 1169130-CF, MPBio) in 1 ml of 0.1 M HCl on an Omni Bead Ruptor Elite bead mill homogenizer (Revvity). After homogenization, the sample was diluted 5-fold. cAMP levels were measured using the Enzo Direct cAMP ELISA kit (ADI-900-066A) following the manufacturer's protocol. Briefly, 50 µl of neutralizing reagent, 100 µl of sample, 50 µL of blue conjugate, and 50 µl of yellow antibody were added to each well of the ELISA plate. The plate was incubated at room temperature for 2 h. After incubation, the wells were washed three times with 400 µl of wash buffer. Then, 200 µl of substrate solution was added to each well and incubated for 1.5 h. Finally, 50 µl of stop solution was added, and absorbance was measured at 405 nm. Protein concentration was determined by bicinchoninic acid assay (Thermo Fisher Scientific). To normalize for protein content, the resulting pmol/ml determinations was divided by the total protein concentration (mg/ml) in each sample and expressed as pmol cAMP/mg protein.

## Human kidney primary cell culture, siRNA transfection and immunostaining

Normal human kidney (NHK Pt.012018 Cortex 3RP, P6) and human ADPKD (ADPKD Pt.071416 -HTB-003: p.Gln1463*, Cyst 1) cells were from PKD-RRC (University of Maryland). Cells were cultured in 1:1 mixture of Lonza REBM Basal Medium (CC-3191) with REGM SingleQuots supplements (CC-4127) and Advanced MEM (Gibco #12492), supplemented with 5% FBS, 2.2% penicillin-streptomycin, 0.6% L-alanyl-glutamine (Glutamax, Life Technologies, #35050061), and 0.03% gentamicin (Life Technologies, #15750060). The siRNAs were predesigned On-target Plus (OTP) siRNA duplexes shown to yield a reduced frequency of off-target effects (Horizon Discovery, previously Dharmacon). The OTP siRNAs are as follows: nontargeting pool, D-001810–10; *ANKMY2*, J-013766–18 and J-013766–19. Cells were cultured on poly-L-Lysine coated coverslips (Corning, 354085) and transfected with siRNA using Lipofectamine RNAiMax (Life Technologies). siRNA transfection methods were as follows: total 200 nM by reverse transfection followed by total 200 nM by forward transfection 24 h after plating. One day later cells were starved with 0.05% FBS for 24 h and fixed for 10 min with freeze-thawed chilled 4% PFA at room temperature [56]. Immunostaining was performed to detect ADCY3 (LS-C204505; LifeSpan BioSciences; 1:1000) and Acetylated tubulin (T6793; Sigma mouse IgG2b, 1:2000) and counterstained with DAPI.

## Quantification and statistical analysis

Cystic area and length of cilia in each mouse genotype was measured using ImageJ software (NIH, Bethesda, MD). For measuring cilia positive for a particular protein, we first identified cilia using acetylated tubulin staining as a ciliary marker. In experiments where LAP-tagged constructs were used to rescue ciliary localization of proteins, we counted cilia only from GFP expressing cells. Next, we carefully counted cilia for the presence of staining corresponding to the protein of interest. All Z-planes containing acetylated tubulin staining were analyzed for staining. We did not use any threshold intensity while counting and all cilia including those showing low intensity staining were regarded as positives. All data in Figures are expressed as mean±SD or SEM. To assess the statistical significance of differences among genotypes we performed one-way ANOVA with Sidak's multiple comparisons tests, or unpaired, two-sided, student's $t$ tests that assumed unequal variances in treatments. Microsoft Excel and GraphPad Prism (GraphPad) were used for statistical analysis. Values of $P < 0.05$ were considered significant.

## Supporting information

**S1 Fig. Early cystogenesis in embryonic-onset PKD is suppressed from lack of ANKMY2. (A)** H&E images of multiple kidneys at P3 used for analyses shown for the designated genotypes. Scale, 1 mm. **(B)** Transcript levels of *Pkd1* and *Ankmy2* in whole kidneys at P3 shown. Horizontal bars, mean.
(TIF)

**S2 Fig. Lack of ANKMY2 has no effect on later cystic burden in embryonic-onset PKD.** H&E images of multiple kidneys at P15 used for analyses shown for the designated genotypes. Scale, 2 mm.
(TIF)

**S3 Fig. Cystogenesis in adult-onset PKD in male mice is suppressed from lack of ANKMY2. (A**) qRT-PCR of whole kidneys showed depletion of *Pkd1* or *Ankmy2* in the respective inducible conditional knockout animals at 5 months. Samples are color coded to show respective levels in the same animal. Horizontal bars, mean. **(B)** H&E images of multiple kidneys from males at 5 months used for analyses shown for the designated genotypes. Scale, 2 mm.
(TIF)

**S4 Fig. Cystogenesis in adult-onset PKD in female mice is not suppressed from lack of ANKMY2. (A)** 2-Kidney to body weight ratios of 5-month-old *Pax8^rtTA^*; *TetO-Cre*; *Pkd1^f/f^* female mice were not significantly different than *Pax8^rtTA^*; *TetO-Cre*; *Pkd1^f/f^*; *Ankmy2^f/f^*. Horizontal bars, mean. **(B)** Cystic index in *Pax8^rtTA^*; *TetO-Cre*; *Pkd1^f/f^* female mice compared to *Pax8^rtTA^*; *TetO-Cre*; *Pkd1^f/f^*; *Ankmy2^f/f^* showed no difference. Horizontal bars, mean. **(C)** Cyst sizes in *Pax8^rtTA^*; *TetO-Cre*; *Pkd1^f/f^* and *Pax8^rtTA^*; *TetO-Cre*; *Pkd1^f/f^*; *Ankmy2^f/f^* female mice were not significantly different from each other in both LTL+ and AQP2+ cysts as counted in Fig 3E - 3F. Superplots of N = 3 kidneys/genotype are shown with cysts from each kidney depicted by smaller identical shapes and averages per kidney by analogous larger shapes. Cystic indices in *Pax8^rtTA^*; *TetO-Cre*; *Pkd1^f/f^* were 56, 44, and 42, whereas that in *Pax8^rtTA^*; *TetO-Cre*; *Pkd1^f/f^*; *Ankmy2^f/f^* were 39, 32, and 23. Data are shown as mean ± SD. ns, not significant using one-way ANOVA with Sidak's multiple comparisons tests. **(D)** H&E images of multiple kidneys from females at 5 months used for analyses shown for the designated genotypes. Scale, 2 mm.
(TIF)

**S5 Fig. Changes in ADCY subcellular localization in mouse and human kidney epithelial cells upon loss of ANKMY2. (A-B)** Surface levels of ADCY5 and ADCY6 in *Ankmy2* ko IMCD3 cells were restored upon exogenous ^HA^ANKMY2 reexpression. ADCY5^LAP^ and ^LAP^ADCY6 were localized to the plasma membrane and secretory pathway in stably expressing IMCD3 FlpIn cell line. Surface levels were reduced in *Ankmy2* ko but restored back upon ^HA^ANKMY2 reexpression, as detected upon performing immunostaining with antibodies against GFP, β-catenin and HA. Scale: 25 µm. **(C)** Sequence of *Ankmy2* exon1 targeted for CRISPR-Cas9 mediated *Ankmy2* knockout in mouse IMCD3 Flp-In cells stably expressing ADCY5^LAP^ or ^LAP^ADCY6. Immunoblotting showing absence of ANKMY2 in the clonal knockout lines are depicted in Fig 5G. **(D)** Cilia lengths measured in IMCD3 and *Ankmy2* ko cells show no significant differences using unpaired, two-sided, student's *t* tests. *n* > 100 cells were counted in each condition. Horizontal bars, mean. **(E-G)** NHK and human ADPKD cells were forward, and reverse transfected with 200 nM *siANKMY2* at an interval of 24 h and after further 24 h was serum starved for 24 h before fixation and immunostaining for ADCY3 and acetylated tubulin (Methods). Scale: 10 µm; insets: 5 µm (E). Transcript levels of *ANKMY2* with respect to *GAPDH* in the same conditions (F) and quantification of ADCY3 positive cilia (%) are shown in (G). *n* > 40 were counted in each condition. Data are shown as mean ± SD. **, P < 0.01; ***, P < 0.001; ns, non-significant using one-way ANOVA with Sidak's multiple comparisons tests.
(TIF)

**S6 Fig. Ciliary length increase from PC1 loss in adult kidney epithelia precedes overt cystogenesis. (A)** Scheme for inducible conditional knockout in kidney nephron epithelia in an adult-onset model of PKD to detect "pre-cystic" tubules. **(B)** H&E and whole mount images of 8-week-old kidneys in control and *Pax8^rtTA^*; *TetO-Cre*; *Pkd1^f/f^* male mice in

C57BL/6J background shown. All kidneys analyzed shown in S7 Fig. **(C)** LTL/AQP2 co-staining in 8 weeks old kidney sections shown. **(D)** 2-kidney to body weight ratios, cystic indices of 8-week-old kidneys in control and *Pax8^rtTA*; *TetO-Cre*; *Pkd1^f/f* male. \*\*\*, P<0.001 with respect to each control using unpaired, two-sided, student's *t* tests; ns, non-significant. **(E)** Cilia lengths were quantified in 8-week-old kidneys in control and *Pax8^rtTA*; *TetO-Cre*; *Pkd1^f/f* males, and super plots are shown for each genotype. Individual cilia lengths from each kidney are shown with identical shapes and averages per kidney are plotted with larger analogous shapes. Horizontal bars, mean of averages from each kidney. Ns, non-significant using one-way ANOVA with Sidak's multiple comparisons tests for all cilia. **(F-I)** Same as in (B-E) except 14-week-old animals of both sexes are shown. \*, P<0.05; \*\*, P<0.01; \*\*\*\*, P<0.0001; ns, non-significant with respect to each control using unpaired, two-sided, student's *t* tests (H) or one-way ANOVA with Sidak's multiple comparisons tests for all cilia (I). (TIF)

**S7 Fig. Tubule dilatation stage kidneys.** H&E images of multiple kidneys in male and female mice quantified in Figs 7D, 7E and S6 are shown for the designated genotypes. Scale: 2 mm. (TIF)

**S1 Data. Numerical data in figures and supplemental figures.** All numerical data in figures and supplemental figures are provided in this spreadsheet. (DOCX)

## Acknowledgments

We thank the transgenic core, molecular pathology, and mouse animal care facility in UT Southwestern. We thank John Shelton for histopathology core support. We also acknowledge the Polycystic Kidney Disease Research Resource Consortium at University of Maryland Baltimore's PKD Center (U54DK126114) for providing the *Pax8^rtTA*; *TetO-Cre*; *Pkd1^f/f* mouse model and support.

## Author contributions

**Conceptualization:** Sun-Hee Hwang, Feng Qian, Saikat Mukhopadhyay.

**Formal analysis:** Sun-Hee Hwang, Saikat Mukhopadhyay.

**Funding acquisition:** Saikat Mukhopadhyay.

**Investigation:** Sun-Hee Hwang, Kyungsuk Choi, Hemant Badgandi, Kevin A White, Yu Xun.

**Methodology:** Sun-Hee Hwang, Kyungsuk Choi, Hemant Badgandi, Kevin A White, Yu Xun, Owen M. Woodward.

**Project administration:** Saikat Mukhopadhyay.

**Resources:** Owen M. Woodward, Feng Qian, Saikat Mukhopadhyay.

**Supervision:** Saikat Mukhopadhyay.

**Validation:** Sun-Hee Hwang, Kyungsuk Choi.

**Visualization:** Sun-Hee Hwang, Kyungsuk Choi.

**Writing – original draft:** Saikat Mukhopadhyay.

**Writing – review & editing:** Sun-Hee Hwang, Kyungsuk Choi, Hemant Badgandi, Yu Xun, Owen M. Woodward, Feng Qian, Saikat Mukhopadhyay.

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
