## [Decision Letter · Decision Letter 0]

13 Oct 2025

PGENETICS-D-25-01008

Lack of adenylyl cyclase targeting to primary cilia suppresses kidney cystogenesis in embryonic- and adult-onset polycystic kidney disease

PLOS Genetics

Dear Dr. Mukhopadhyay,

Thank you for submitting your manuscript to PLOS Genetics. After careful consideration, we feel that it has merit but does not fully meet PLOS Genetics's publication criteria as it currently stands. Therefore, we invite you to submit a revised version of the manuscript that addresses the points raised during the review process.

Please submit your revised manuscript within 60 days Dec 12 2025 11:59PM. If you will need more time than this to complete your revisions, please reply to this message or contact the journal office at plosgenetics@plos.org. Please include the following items when submitting your revised manuscript:

We look forward to receiving your revised manuscript.

Kind regards,

Gregory J. Pazour, PhD

Academic Editor

PLOS Genetics

Fengwei Yu

Section Editor

PLOS Genetics

Aimée Dudley

Editor-in-Chief

PLOS Genetics

Anne Goriely

Editor-in-Chief

PLOS Genetics

**Additional Editor Comments:**

Saikat,

Thank you for submitting this work to Plos Genetics. As you can see from the comments below, both reviewers found your study interesting and important. However, both had serious concerns about interpretation of the data that will need to be addressed before acceptance. It appears from the comments that more experimental work will likely be needed to fill in the gaps.

Greg

**Journal Requirements:**

- TM on page: 24.

4) We notice that your supplementary Figures are included in the manuscript file. Please remove them and upload them with the file type 'Supporting Information'. Please ensure that each Supporting Information file has a legend listed in the manuscript after the references list.

Potential Copyright Issues:

i) Please confirm (a) that you are the photographer of 2B, 3A, and and 6A, or (b) provide written permission from the photographer to publish the photo(s) under our CC BY 4.0 license.

ii) Figure 8. Please confirm whether you drew the images / clip-art within the figure panels by hand. If you did not draw the images, please provide (a) a link to the source of the images or icons and their license / terms of use; or (b) written permission from the copyright holder to publish the images or icons under our CC BY 4.0 license. Alternatively, you may replace the images with open source alternatives. See these open source resources you may use to replace images / clip-art:

6) In the online submission form, you indicated that "Mice strains described will be made available to other researchers upon request. All expression plasmids described are available from the authors." All PLOS journals now require all data underlying the findings described in their manuscript to be freely available to other researchers, either

1. In a public repository

2. Within the manuscript itself

3. Uploaded as supplementary information.

7) Please amend your detailed Financial Disclosure statement. This is published with the article. It must therefore be completed in full sentences and contain the exact wording you wish to be published.

8) We note that your Data Availability Statement is currently as follows:"All data are available in the main text or the supplementary materials." Please confirm at this time whether or not your submission contains all raw data required to replicate the results of your study. Authors must share the “minimal data set” for their submission.

PLOS defines the minimal data set to consist of the data required to replicate all study findings reported in the article, as well as related metadata and methods (https://journals.plos.org/plosone/s/data-availability#loc-minimal-data-set-definition). For example, authors should submit the following data: -

-The values behind the means,  standard deviations and other measures reported; 

-The values used to build graphs; -

-The points extracted from images for analysis.

Authors do not need to submit their entire data set if only a portion of the data was used in the reported study. If your submission does not contain these data, please either upload them as Supporting Information files or deposit them to a stable, public repository and provide us with the relevant URLs, DOIs, or accession numbers For a list of recommended repositories, please see "https://journals.plos.org/plosone/s/recommended-repositories"

9) Please confirm whether your study includes live participants. If your study involves live participants, please insert an Ethics Statement at the beginning of your Methods section, under a subheading 'Ethics Statement'. It must include:

i) The full name(s) of the Institutional Review Board(s) or Ethics Committee(s)

ii) The approval number(s), or a statement that approval was granted by the named board(s).

**Reviewers' comments:**

Reviewer's Responses to Questions

Reviewer #1: The manuscript by Hwang et al. entitled “Lack of adenylyl cyclase targeting to primary cilia suppresses kidney cystogenesis in embryonic- and adult-onset polycystic kidney disease” investigates the role of ciliary AC/cAMP signaling controlled by ANKMY2 in cystogenesis. The authors generated kidney-specific Ankmy2/Pkd1 knockout mice, demonstrating that loss of ANKMY2 suppressed early postnatal cystogenesis and prolonged survival. Mechanistically, the authors attribute this to inhibition of the ciliary length increase, which is generally observed in ADPKD models. The authors conclude that ciliary ACs promote cystogenesis and that these mechanisms are distinct from the cAMP-dependent mechanisms in the cytosol. The results are novel and complementary to other results in the field, demonstrating that ciliary cAMP is the main driver of cystogenesis. However, the conceptual advance in terms of mechanistic insights needs to be further improved. Thus, the following points need to be addressed:

1. Figure 2/5: The authors use pCREB as a read-out for total cAMP production in the cells, which is only a proxy. This should be analyzed by sorting epithelial cells and determine total cAMP levels using ELISA, as shown in Fig. 6. The experiment in Figure 6 needs to be done using epithelial cells only if they have been performed using whole tissue. There are easy ways to enrich epithelial cells using MACS (Epcam).

2. In addition, there is also literature demonstrating that CREB shuttles through the cilium, where it gets activated by ciliary cAMP/PKA, evoking a gene expression program that is distinct from the cytoplasmic cAMP-dependent program (Hansen et al., 2022). As the authors are analyzing the action of ACs in the cilium, the subcellular localization of pCREB, including the cilium, would need to be analyzed.

3. Figure 2/5: Another major point that is lacking is that no direct measurement for the change of cAMP/PKA signaling in the cilium have been performed. This can be done in isolated cells using biosensors for cAMP and PKA. As biosensor imaging might be difficult in primary cells, the authors should perform those measurements in the generated IMCD-3 cells lacking Ankmy2. These outcome of these experiments could also be correlated with cilia length.

4. Information about the statistical tests is missing in some of the legends

5. All graphs should be presented as mean + S.D. and individual values.

Reviewer #2: This is an interesting study, where it is suggested that ciliary ADCY5/6-mediated cAMP signaling is downstream of Pkd1-induced ADPKD. The study includes both mouse and cell culture work and if revised, it will provide important mechanistic refinements about the role of the cAMP pathway in ADPKD. However, there are conceptual gaps that need to be addressed. Deletion of PC1 or PC2 leads to an increase in total cellular cAMP levels, as is demonstrated here and in many other publications. This means that PC1 and PC2 function not only in cilia but also in several other places to cause a cellular increase in cAMP upon their deletions. Thus, their contribution to the regulation of cAMP is way more global (at the cellular level) than cAMP in the cilium, which makes sense as they are not there to just prevent cystogenesis. Whether the ciliary cAMP pool is the relevant pool to prevent cystogenesis, however, is an important question and the subject of the current study. This is viewed as a strength of the manuscript. A major general weakness is that conclusions are drawn mostly based on conjecture and not direct evidence. For example, there is no direct evidence that deletion of Ankmy2 leads to the specific depletion of ADCY5/6 in cilia in tissues, no demonstration that there is abnormal cAMP accumulation in Pkd1-null cilia, which is reduced by the depletion of Ankmy2 (even in cell culture), lack of consideration of effects of Ankmy2 on other ADCYs such as ADCY3, and lack of mechanisms of how intraciliary cAMP causes ciliary elongation. While it is not expected that all these issues be addressed in a single manuscript, some key points (see below in specific points) should be addressed to strengthen the conclusions. The ameliorating phenotypes of Adcy5 or Adcy6 in double mutant mice are extremely weak (PMC5794572, PMC3904559), so it is difficult to see how their partial depletion from cilia can have strong rescuing effects (Figure 3B), especially when deletion of Pkd1 causes a global (cellular) increase in cAMP. Finally, previous work has shown that there is an upregulation of cellular pCREB in collecting ducts, but not in proximal tubules, despite the presence of proximal tubule cysts (PMC3758452). Yet the authors show rescue in late models in both proximal tubules and collecting ducts. It is possible that if the ciliary cAMP pool is relevant to cystogenesis and loss of Pkd1 induces an increase in just the ciliary levels of cAMP in proximal tubules, these cAMP changes may be difficult to capture by looking at total cAMP levels. However, this needs to be shown. In this regard, accumulation of pCREB in the cilia of both kidney segments may be more telling than total cellular pCREB levels.

Specific points

1) Demonstration that depletion of one or more of the ADCYs in cilia in kidney tissues in Ankmy2-null kidneys will be needed. It is understandable that existing antibodies for ADCY5/6 are not reliable but given the lack of clear understanding of how exactly ANKMY2 affects the trafficking of these ADCYs in the cilia, it is an essential point. In this regard, how about ADCY3? Antibodies for this isoform should work on cilia. Whether there is Ca2+ dependance (upon PC1/2 deletion) or not of the cAMP regulation in the cilium is a separate question and if not, there could be alternative explanations for this. Any surrogate assay based on the purported cellular function of ANKMY2 affecting ADCY maturation would suffice. Anything substantial to increase confidence that the presence/function of these ADCYs is reduced specifically in the cilium in Ankmy2-null kidneys would suffice.

2) Demonstration that there is accumulation of pCREB in the cilium or anything else that would reflect increased cAMP in the cilium in Pkd1-null cells that is reversed in double mutant mice would be needed. Total pCREB level is in disagreement with effects on proximal tubules and ciliary length is not an established readout of increased ciliary cAMP. Experiments in proximal tubules may be more relevant than IMCD cells in this case and could help the authors make their case stronger.

3) What is interpretation that the rescuing effect is lost at P15 (Figure 2)? The deterioration of the phenotype from P3 to P15 argues for a dual effect on Ankmy2/cAMP on cystogenesis.

4) Is ANKMY2 expressed in Hox7+ and Pax8+ cells?

5) Sexual dimorphism is suggested as a possible explanation for the sex differences, but this is not tested.

6) Experiments in cell culture are difficult to interpret (Figure 5 and S5). Levels of glycosylation of ADCY5/6 are reduced in ANKMY2 KO cells, but how does this affect ciliary expression versus plasma membrane expression? The data are more consistent with a global effect of ANKMY2 on maturation and processing of ADCY5/6 towards both the plasma and cilia membrane, unless the authors suggest that ANKMY2 promotes/facilitates the lateral movement of these ADCYs from the plasma membrane to the cilium via the transition zone or similar structures that control ciliary entry of transmembrane proteins. If so, this needs to be shown. The conclusion in lines 298-301, p. 12 is unsupported. How about cilia length in these cells?

7) Figure 7E. cAMP levels represent total levels. These data are confusing. There is a positive correlation between cAMP levels and cystic index in single and double mutants. How can it possible, if Ankmy2 only affects the ciliary pool that should generate undetectable levels of cAMP? I’d expect positive correlation in single ko male mice, but no correlation of total cAMP and cystic index in male dko mice.

**Have all data underlying the figures and results presented in the manuscript been provided?**

Reviewer #1: Yes

Reviewer #2: Yes

PLOS authors have the option to publish the peer review history of their article (what does this mean? ). If published, this will include your full peer review and any attached files.

**Do you want your identity to be public for this peer review?** For information about this choice, including consent withdrawal, please see our Privacy Policy .

Reviewer #1: No

Reviewer #2: No

**Figure resubmission:**

**Reproducibility:**



---

## [Editor Report · Decision Letter 1]

17 Dec 2025

PGENETICS-D-25-01008R1

Lack of ANKMY2 suppresses kidney cystogenesis in embryonic- and adult-onset polycystic kidney disease

PLOS Genetics

Dear Dr. Mukhopadhyay,

I have carefully gone through your manuscript and feel that you have adequately addressed most of the reviewer concerns by either new work or by adjusting the text to better describe what was observed. The only concern that I have is that the request to describe the statistical tests in the figure legends was not adequately addressed. The materials and methods list several tests and say that the figures show either mean+/-SD or mean+/-SEM. However. the figure legends lack information about how the data in the figure was analyzed. The figure legends should clearly state the test used to obtain the P values listed and state whether the error bars (if you have them) are SEM or SD.

Thank you for submitting this work to PLOS Genetics!

Please submit your revised manuscript within by Jan 16 2026 11:59PM. If you will need more time than this to complete your revisions, please reply to this message or contact the journal office at plosgenetics@plos.org. Please include the following items when submitting your revised manuscript:

We look forward to receiving your revised manuscript.

Kind regards,

Gregory J. Pazour, PhD

Academic Editor

PLOS Genetics

Fengwei Yu

Section Editor

PLOS Genetics

Aimée Dudley

Editor-in-Chief

PLOS Genetics

Anne Goriely

Editor-in-Chief

PLOS Genetics

**Journal Requirements:**

1) Please amend your detailed Financial Disclosure statement. This is published with the article. It must therefore be completed in full sentences and contain the exact wording you wish to be published.

1) State what role the funders took in the study. If the funders had no role in your study, please state: "The funders had no role in study design, data collection and analysis, decision to publish, or preparation of the manuscript.".

2) Please update the Data Availability Statement provided in the online submission form and ensure that it matches the one stated in the manuscript.

**Figure resubmission:**
---

## [Editor Report · Decision Letter 2]

21 Dec 2025

Dear Dr Mukhopadhyay,

We are pleased to inform you that your manuscript entitled "Lack of ANKMY2 suppresses kidney cystogenesis in embryonic- and adult-onset polycystic kidney disease" has been editorially accepted for publication in PLOS Genetics. Congratulations!

Yours sincerely,

Gregory J. Pazour, PhD

Academic Editor

PLOS Genetics

Fengwei Yu

Section Editor

PLOS Genetics

Aimée Dudley

Editor-in-Chief

PLOS Genetics

Anne Goriely

Editor-in-Chief

PLOS Genetics

BlueSky: @plos.bsky.social

Comments from the reviewers (if applicable):

**Data Deposition**

http://datadryad.org/submit?journalID=pgenetics&manu=PGENETICS-D-25-01008R2

**Press Queries**

---

## [Editor Report · Acceptance letter]

PGENETICS-D-25-01008R2

Lack of ANKMY2 suppresses kidney cystogenesis in embryonic- and adult-onset polycystic kidney disease

Dear Dr Mukhopadhyay,

We are pleased to inform you that your manuscript entitled "Lack of ANKMY2 suppresses kidney cystogenesis in embryonic- and adult-onset polycystic kidney disease" has been formally accepted for publication in PLOS Genetics! Your manuscript is now with our production department and you will be notified of the publication date in due course.

With kind regards,

Anita Estes

PLOS Genetics

On behalf of:
